# A Targeted Learning Framework for Policy Evaluation with Unobserved Network Interference

## Abstract

Estimating causal effects under network interference is a fundamental yet challenging task, especially when the network structure is represented as multiple layers or multiple views. In this paper, we consider a heterogeneous network setting, where the ties from different views of the network might achieve varying levels of interference. Meanwhile, dependence among units is allowed, due to information transmission among network ties and latent traits among units sharing ties (i.e., latent dependency). To the best of our knowledge, this setting has not been studied in literature yet. We propose a novel framework that conducts doubly robust estimation on heterogeneous networks with latent dependency. Our approach relies on a new identification strategy and integrates it with targeted maximum likelihood estimation for robust causal effect estimation from observational data. Crucially, our approach remains valid even when the outcome prediction model or data-generating process is misspecified. It also supports counterfactual inference under hypothetical interventions defined on constructed exposure summaries, using only the observed network structure. Experiments on both synthetic and real-world networks show that our approach consistently outperforms existing baselines and can provide robust estimation towards different intervention policies.

## 1 Introduction

Estimating causal effects under network interference is fundamental to evidence-based policy-making in social science, public health, and economics. A central challenge is that the outcome of a unit may depend not only on its own treatment, but also on the treatments received by its neighbours (van der Laan, 2014), violating the classical SUTVA assumption. This phenomenon, known as interference, is pervasive in real-world networks: vaccination campaigns create herd immunity, advertising interventions create spillover demand, and job-training programmes affect peers' employment outcomes. A second, underappreciated challenge is that real-world interference networks are rarely homogeneous or fully observed. Units may be connected via multiple types of relationships—for example, co-viewing and co-purchasing links on an e-commerce platform (He & McAuley, 2016)—and the true interference network is often only partially observed or must be approximated from auxiliary data. These two challenges—heterogeneous multi-view structure and partial observability—are rarely addressed jointly in the causal inference literature.

A common approach to model interference is to define exposure as the proportion of treated neighbors (Forastiere et al., 2022), which allows the estimation of spillover effects, i.e., the effect of varying neighborhood exposure on a unit's outcome. In contrast to modeling exposure solely as a proportion, Toulis & Kao (2013) introduced a framework that defines exposure through the exact $k$-level exposure, i.e., the potential outcome of a node when exactly $k$ of its neighbors are treated. Another way to model network interference involves modeling the transmission of information through network ties using graph neural networks (GNN) based methods (Tchetgen et al., 2019; Ma et al., 2021; 2022; Ma & Tresp, 2021) or directly modeling the confounders to address latent network dependency (Guo et al., 2019; Ma et al., 2021; Zhang et al., 2017).

However, real-world networks may be more complex. One complexity comes from the heterogeneous relationships among units. For example, in Amazon dataset(He & McAuley, 2016), ties may represent different types of interaction such as co-viewing and co-purchasing, and the interference may vary between different relational aspects. Lin et al. (2023) proposed HINITE, which aims to estimate the effect of treatment

in a heterogeneous setting by aggregating the covariate information on different aspects of the networks. Related efforts also include modeling heterogeneity in peer influence using structured graph-based causal models (Adhikari & Zheleva, 2025), and modeling heterogeneity in causal pathways across subpopulations with different moderator variables (Watson et al., 2023). These methods work well when the outcome model is correctly specified for the data, however, they are usually sensitive to model misspecification.

To avoid failure caused by misspecification between the data generation process and the modeling process, the estimation of maximum likelihood is a general framework to construct an objective estimate (Van Der Laan & Rubin, 2006; Van der Laan et al., 2011). This method has been widely used in various settings (van der Laan & Gruber, 2012; Kreif et al., 2017; Chen et al., 2023; Balzer et al., 2019). Shi et al. (2019) proposed targeted regularization for binary treatment effect estimation, while Nie et al. (2021) proposed an end-to-end targeted maximum likelihood estimation (TMLE) framework to estimate the continuous treatment effect. Based on previous work, Chen et al. (2024) proposed T-Net, which models the potential outcome as a function of unit treatment and neighborhood exposure in a homogeneous network setting. In this line of research, only one method (i.e., T-Net) is designed for causal inference on networked data, but it still cannot account for heterogeneity and latent dependency in the network structure.

To close the gap, we propose a novel framework, named **M**ulti-**v**iew **d**oubly **r**obust estimator (**Mvdr**), for causal inference under heterogeneous network interference. The challenge of heterogeneity remains valid even when only a single partially observed network is available. Our goal is to estimate the average expected potential outcome under different hypothetical intervention policies over an observed network with heterogeneous ties. Our approach models interference via multi-view representations of the network and integrates these with a TMLE procedure. In this way, the learned estimators are doubly robust, i.e., they remain consistent if either the outcome model or the exposure model is correctly specified. Our Mvdr framework enables robust estimation of average potential outcomes under a variety of hypothetical interventions, such as static, dynamic, or stochastic interventions. The major contributions of this paper can be summarized as follows:

1. We define policy-value estimands on constructed multi-view exposures and show they are identified for any suitable aggregation (Lemma 4.1), so the TMLE inherits double robustness even when the summary functions are learned from data rather than pre-specified. This generalizes prior TMLE-based network estimators, which require a fixed, known exposure mapping. We further prove a bias bound (Proposition 4.1) relating these estimands to effects on the true interference structure.

2. We propose Mvdr, a multi-view doubly robust estimator that integrates graph neural network-based summary learning with TMLE. Unlike existing methods that either assume a correctly specified outcome model or a single fixed network, Mvdr simultaneously handles: (i) heterogeneous network structure via multi-view GCN representations, (ii) latent unit-level dependence via a local dependence assumption, and (iii) model misspecification via the TMLE perturbation step. To our knowledge, this is the first framework to jointly address all three challenges.

3. We design new evaluation settings with dynamic and stochastic interventions beyond the standard static setting, construct a new semi-synthetic DBLP dataset, and conduct extensive experiments demonstrating that Mvdr consistently outperforms baselines, particularly under dynamic and stochastic policies where exposure mechanism misspecification is most consequential.

The rest of this paper is organized as follows. In Section 2, we briefly present related work on causal inference under network interference, complex interference structure, and model misspecification. Section 3 introduces preliminary and the problem settings. Section 4 presents our theoretical analysis. In Section 5, we introduce the technical details of the proposed Mvdr framework. Section 6 provides experimental results with discussions, and Section 7 concludes this paper.

## 2  Related Works

**Causal Inference under Network Interference.** Classical causal inference frameworks assume that there is no interference between units, but real-world settings often violate this assumption. The foundational work in (Halloran & Struchiner, 1995; Sobel, 2006; Hudgens & Halloran, 2008) formalized potential outcomes

under interference. Manski (2013) and Aronow & Samii (2017) introduced exposure mappings, summarizing neighbor treatments into interpretable variables. Building on this, Karwa & Airoldi (2018) investigated the consequences of misspecified exposure mappings in randomized experiments, while Leung (2022) studied approximate neighborhood interference and provided inference methods under specification error. More recently, Ogburn et al. (2022) developed a semiparametric TMLE framework for causal inference on a single network, allowing dependence to grow with network degree. Sävje (2024) emphasized the distinction between exposures as effect definitions versus structural assumptions, proposing expected exposure effects that remain meaningful under misspecification.

**Complex Interference Structure.** This motivates the construction of proxy networks based on observed covariates. Egami (2021) considered spillover effects when the interference network is only partially observed, while Li & Wager (2022) studied network interference under latent graphon models and equilibrium settings. Parallel work in graph mining has explored multiview and multiplex networks, where multiple graphs encode different relations among the same nodes. Such methods aggregate heterogeneous views through attention or weighting. These ideas have not yet been fully integrated into causal inference. Our work is motivated by the observation that units may be dependent not only on direct social ties, but also on shared environments, socioeconomic status, or latent similarities. Thus, constructing multiple similarity-based networks from observed covariates provides a flexible way to capture such heterogeneity.

**Model Misspecification.** A growing literature addresses robustness when models meet with misspecification with data generating process. Leung (2022) proposed alternative estimands for approximate interference, while Sävje (2024) showed that conventional estimators remain unbiased for expected exposure effects under weakly dependent specification errors. Chao et al. (2025) studied misclassified or surrogate networks, introducing methods to correct bias using validation data. Similarly, Hoshino et al. (2024) considered causal inference with noncompliance and unknown network structures. These works highlight that perfect knowledge of the interference graph is rarely available, and meaningful causal estimands can still be defined under approximation. Our approach draws directly on these insights. By treating similarity-based graphs as proxies rather than true causal pathways, we target expected exposure effects that remain interpretable and estimable even if the constructed networks do not fully align with the underlying social process.

## 3 Preliminary

In this section, we first introduce the notations used in this paper and then describe the preliminary of our work, such as the structural equation model and assumptions. Finally, we present the problem setting and introduce the causal estimands of interest in detail.

### 3.1 Notations

Let $x_i \in \mathbb{R}^d$ denote the covariates of unit $i$, $t_i \in \{0, 1\}$ the binary treatment assignment, and $y_i \in \mathbb{R}$ the observed outcome for unit $i$ under treatment $t_i$. We write $X = (x_1, \ldots, x_n)^\top \in \mathbb{R}^{n \times d}$, $T = (t_1, \ldots, t_n)^\top \in \{0, 1\}^n$, and $Y = (y_1, \ldots, y_n)^\top \in \mathbb{R}^n$ for the covariates, treatments, and outcomes of all $n$ units.

**Network views.** The interference structure among units is represented by $K$ *pre-treatment views*, with symmetric adjacency matrices $A^{(1)}, \ldots, A^{(K)} \in \{0, 1\}^{n \times n}$, where $A_{ij}^{(k)} = 1$ if units $i$ and $j$ are tied in view $k$. Each view captures a specific type of relationship and may be either *directly observed* (e.g., a co-authorship or social network) or *constructed* from baseline covariates (e.g., a $k$-nearest-neighbour similarity graph). We denote by $N_i^{(k)} = \{j : A_{ij}^{(k)} = 1\}$ the neighbourhood of unit $i$ in view $k$ and by $d_i^{(k)} = |N_i^{(k)}|$ its degree.

**Union support graph.** We use the *union support graph* $\tilde{A} \in \{0, 1\}^{n \times n}$, defined by $\tilde{A}_{ij} = 1$ if and only if $A_{ij}^{(k)} = 1$ for at least one view $k \in \{1, \ldots, K\}$. We write $\tilde{N}_i = \{j : \tilde{A}_{ij} = 1\}$ and let $K_{\max,n} := \max_i |\tilde{N}_i|$ denote its maximum degree. We emphasize that the views (and hence $\tilde{A}$) are working representations of the interference structure, not an assertion that they coincide with the true channels through which interference operates. A table of all notations is provided in Appendix A.

### 3.2 Structural Equation Model

We assume that the observed data arise from a single network of size $n$. Units may exhibit latent variable dependence: unobserved traits shared by related units can induce correlations in their covariates, treatments, or outcomes (Shalizi & Thomas, 2011). Such dependence is typically stronger between units that are close

in the underlying relational structure — which need not be fully captured by any single observed adjacency matrix; this motivates representing the interference structure through multiple views, as introduced in Section 3.1.

To formalize the data generating process, we adopt the structural equation model (SEM) framework (Pearl, 2012) and posit that the data are generated by sequentially evaluating

$$
\begin{aligned}
&X_i = f_X[\epsilon_{X_i}], \ i = 1, \dots, n \\
&T_i = f_T[\{X_j : \tilde{A}_{ij} = 1\}, \epsilon_{T_i}], \ i = 1, \dots, n \\
&Y_i = f_Y[\{T_j : \tilde{A}_{ij} = 1\}, \{X_j : \tilde{A}_{ij} = 1\}, \epsilon_{Y_i}], \ i = 1, \dots, n
\end{aligned}
\tag{1}
$$

where the neighbourhoods are taken with respect to the union support graph $\tilde{A}$ of Section 3.1, $f_X$, $f_T$, and $f_Y$ are unknown and unspecified functions that may depend on the unit $i$, and $\epsilon_{X_i}$, $\epsilon_{T_i}$, $\epsilon_{Y_i}$ are exogenous, unobserved errors. Errors of the same type may be correlated across units (the precise dependence structure is restricted by Assumption 4). We emphasize that Eq. (1) is a *working* structural model with respect to the constructed views: it posits that interference is transmitted along ties present in at least one view. Proposition 4.1 quantifies the consequences when the true interference operates through a different structure.

Based on this model, we define view-specific summaries

$$
W_i^{(k)} = s_X^{(k)}\big(\{X_j : A_{ij}^{(k)} = 1\}\big), \qquad V_i^{(k)} = s_T^{(k)}\big(\{T_j : A_{ij}^{(k)} = 1\}\big), \qquad k = 1, \dots, K,
$$

and aggregate them into

$$
W_i = \mathrm{Agg}_W\big(W_i^{(1)}, \dots, W_i^{(K)}\big), \qquad V_i = \mathrm{Agg}_V\big(V_i^{(1)}, \dots, V_i^{(K)}\big),
$$

where $s_X^{(k)}, s_T^{(k)}$ and the aggregators $\mathrm{Agg}_W, \mathrm{Agg}_V$ are Borel-measurable maps. The model in Eq. (1) can then be rewritten as

$$
\begin{aligned}
&X_i = f_X[\epsilon_{X_i}], \ i = 1, \dots, n \\
&T_i = f_T[W_i, \epsilon_{T_i}], \ i = 1, \dots, n \\
&Y_i = f_Y[V_i, W_i, \epsilon_{Y_i}], \ i = 1, \dots, n
\end{aligned}
\tag{2}
$$

Eq. (2) states that the outcome $Y_i$ depends on $(\mathbf{X}, \mathbf{T})$ only through the aggregated summaries $(W_i, V_i)$: the covariates of $i$ and its neighbours enter via $W_i$, and the treatments of $i$ and its neighbours enter via $V_i$.

### 3.3 Assumptions

Based on the SEM model, we make the following assumptions:

**Assumption 1 (Pre-treatment views):** Each view $A^{(k)}$, $k = 1, \dots, K$, is fixed prior to treatment assignment: it is either a directly observed pre-treatment network or a function $G_k(X)$ of baseline covariates. In either case, edges do not depend on the treatments $T$ or outcomes $Y$.

**Assumption 2 (Ignorability):** The error vectors $(\epsilon_{T_1}, \dots, \epsilon_{T_n})$, $(\epsilon_{Y_1}, \dots, \epsilon_{Y_n})$, and $(\epsilon_{X_1}, \dots, \epsilon_{X_n})$ are mutually independent.

**Assumption 3 (Exchangeability):** Within each error type, the errors $\epsilon_{T_1}, \dots, \epsilon_{T_n}$ are identically distributed, and likewise for $\epsilon_{X_1}, \dots, \epsilon_{X_n}$ and $\epsilon_{Y_1}, \dots, \epsilon_{Y_n}$.

**Assumption 4 (Local Latent Dependence Structure):** Within each error type, the errors of units $i$ and $j$ (e.g., $\epsilon_{X_i}$ and $\epsilon_{X_j}$, and analogously for $\epsilon_T$ and $\epsilon_Y$) may be dependent only if $i$ and $j$ share a neighbour in the union support graph, i.e., only if $\exists\, u$ such that $\tilde{A}_{iu} = \tilde{A}_{uj} = 1$; otherwise they are independent.

**Assumption 5 (Positivity):** For all $i$, the conditional law $h_i(v \mid w) = P(V_i = v \mid W_i = w)$ (a probability mass function when $V$ is discrete, or a density when $V$ contains a continuous component) satisfies $h_i(v \mid w) > 0$ for all $w$ in the support of $W$ and all $v$ in the support of $V$.

**Assumption 6 (Exposure sufficiency):** $Y_i = f_Y[V_i, W_i, \epsilon_{Y_i}]$, i.e., the constructed summaries $(V_i, W_i)$ are sufficient statistics of $(\{T_j\}, \{X_j\})$ for the outcome. Assumptions 1–5 are maintained throughout; Assumption 6 is invoked only where stated (Proposition 4.1).

The SEM in Eq. (2) encodes the assumption that the summaries $(W_i, V_i)$ suffice to control for confounding of the exposure–outcome relationship — a network analogue of the conditional ignorability assumption standard in i.i.d. settings, made precise by Assumptions 2–3. Assumption 1 rules out post-treatment bias and ensures that all views, and hence $\tilde{A}$, can be regarded as fixed prior to interventions. Assumption 4 permits latent dependence among units, but restricts it to pairs within two hops in $\tilde{A}$; the two-hop radius arises because the summaries $(W_i, V_i)$ aggregate one-hop neighbourhoods, so units sharing a neighbour have overlapping summaries. Assumptions 1–3 and 5 are used for identification (Lemma 4.1), with Assumption 5 additionally ensuring the conditional expectations defining our estimands are well-defined, and Assumption 4 is used only for the asymptotic theory (Theorem 4.1).

### 3.4 Problem Setting and Estimands

Our goal is to estimate the causal effects of hypothetical intervention policy $T^*$ applied to the network. We can construct an average expected outcome that evaluates the causal effect of an observed network receiving certain interventions, based on the conditional-independence structure encoded in Eq. (2) (Van der Laan, 2014):

$$E[\bar{Y}_n^*] = \frac{1}{n}\sum_{i=1}^n E[Y_i^*] = \frac{1}{n}\sum_{i=1}^n E[m(s_{T,i}(T^*), s_{X,i}(x))] = \frac{1}{n}\sum_{i=1}^n \sum_w m(v_i^*, w)h_i^*(v_i^*.w), \qquad (3)$$

where $h$ refers to the conditional distributions of $W$ and $V$: $h_i(v \mid w) = P(V_i = v \mid W_i = w)$ and $h_i(v, w) = P(V_i = v, W_i = w)$. $m(v, w) = \sum_y y p_Y(y \mid v, w)$ is the conditional expectation of the outcome $Y$ given $V = v$ and $W = w$.

Our target parameter is thus defined on the constructed exposure summaries $(V, W)$: it is the expected outcome under interventions on $V$, an expected exposure effect in the sense of Sävje (2024). It is identified from $P(W, V, Y)$ under Assumptions 1–3 and 5 (Lemma 4.1). Proposition 4.1 relates it to the effect defined on the true interference structure.

The causal estimands of interest are identified by functionals of the observed data distribution $P(X, T, Y)$, which depends on $n$ and the views $A^{(1)}, \dots, A^{(K)}$ in our assumption — equivalently, since $(W, V)$ are measurable functions of $(X, T, A^{(1)}, \dots, A^{(K)})$, of $P(W, V, Y)$. It is a parameter of the observed data distribution for a network of size $n$, that is, a parameter of the data generating distribution that gave rise to the data at hand. It is an unknown parameter rather than an observed quantity, because the data we observe comprise a single random draw from $P(X, T, Y)$. Our estimand thus depends on two nuisance parameters: an outcome regression $m$, a function of the treatment summary $V$ and covariate summary $W$ that predicts the potential outcome at a given $(v_i, w_i)$, and a conditional density $h$ that gives the probability of a realization $(v_i, w_i)$.

## 4 Theoretical Analysis

### 4.1 Efficient Influence Curve

The efficient influence curve is a key ingredient in semi-parametric efficient estimation, which defines the linear approximation of any efficient and regular asymptotically linear estimator. Therefore, it provides an asymptotic bound for the variance of all regular asymptotically linear estimations. Under the assumptions made in Section 3.3, the influence function for the causal estimand of interest, which is denoted as $\psi_n$, can be written as:

$$D_N(o) = \frac{1}{n}\sum_{i=1}^n (E[m(V_i^*, W_i) \mid X = x] - \psi_n + \frac{\bar{h}^*(v_i, w_i)}{\bar{h}(v_i, w_i)}\{y_i - m(v_i, w_i)\}), \qquad (4)$$

where $\bar{h}(v_i, w_i) = \frac{1}{n}\sum_{j=1}^n h_j(v_i, w_i)$ and $\bar{h}^*(v_i, w_i) = \frac{1}{n}\sum_{j=1}^n h_j^*(v_i, w_i)$. The target parameter $\psi_n$ is defined as the unique value satisfying $\mathbb{E}[D_N(o)] = 0$; $D_N(o)$ is the estimating equation whose solution identifies $\psi_n$.

Estimating equations for the parameters indexing a working model for $m$ and $h$ are stacked with the influence function estimating equation for $\psi_n$ (Kennedy, 2016). The key idea of the targeted maximum likelihood estimation (TMLE) is to make the influence function equal to 0. The TMLE procedure consists of four steps: **Initial estimation**, which fits working models $\hat{m}(v, w)$ and $\hat{h}(v \mid w)$ from the observed data $(W, V, Y)$;

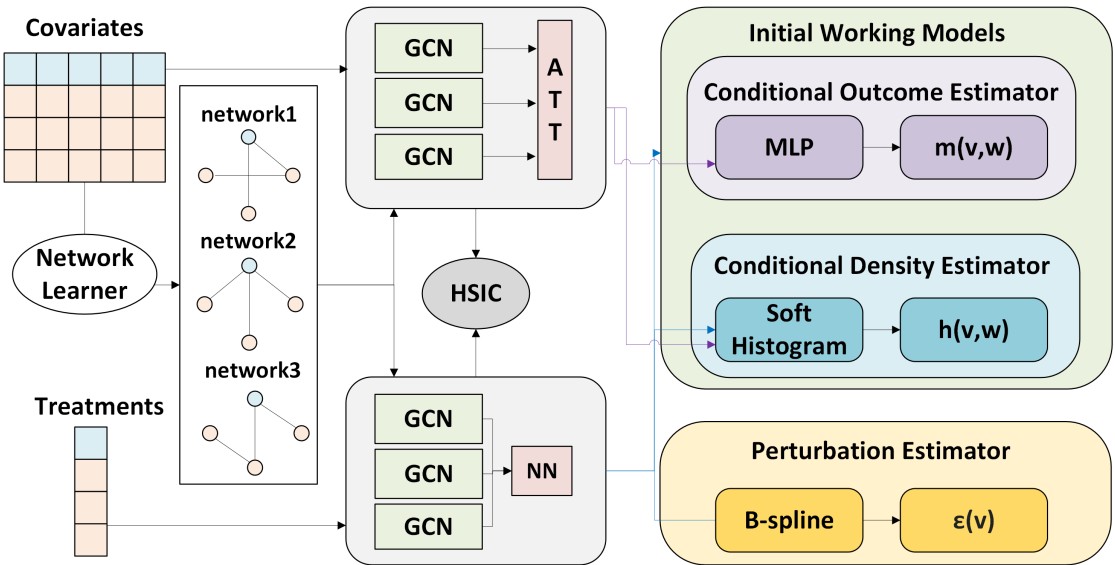

Figure 1: Illustration of our Mvdr framework. Mvdr contains four modules. **Network Learner:** Constructs $K$ view-specific networks from covariates using KNN or other methods. We use $K = 3$ in this figure for simplification. The **representation module** learns the summary function of **X** and aggregate the information from heterogeneous networks, the observed one, i.e., *network1*, and constructed ones (*network2* and *network3*). The **initial working model** contains one conditional outcome estimator $m$ and a conditional density estimator $h$. The **perturbation estimator** is used to conduct TMLE.

**clever covariate computation**, which computes the importance weight $H_i = \bar{h}^*(v_i, w_i)/\bar{h}(v_i, w_i)$; **targeted update** which fits the perturbation parameter $\epsilon(v)$ by minimising the influence function loss and finally **policy value estimation.** The estimator is doubly robust: Eq. (4) has expectation zero if either $\hat{m}$ or $\hat{h}$ is correctly specified, as long as the other remains bounded.

Let $O = (W, V, Y)$ denote the observed variables, where $W$ and $V$ are aggregated summaries constructed from multi-view networks. Write $m(v, w) = \mathbb{E}[Y \mid V = v, W = w]$, $\bar{h}(v \mid w) = \Pr(V = v \mid W = w)$ (or density if $V$ is continuous), and $p_W$ the marginal law of $W$. We consider two common targets.

**Target A (Static neighbor intervention).** For a fixed $v_0$ in the support of $V$, define

$$\psi \;=\; \Psi(P) \;=\; \mathbb{E}\big\{\, m(v_0, W) \,\big\} \;=\; \int m(v_0, w)\, p_W(dw).$$

Under the nonparametric model $M = M_m \times M_h \times M_{p_W}$, the efficient influence function (EIF) is:

$$D^*(O) \;=\; \frac{\mathbb{1}\{V = v_0\}}{\bar{h}(v_0 \mid W)} \big\{ Y - m(v_0, W) \big\} \;+\; m(v_0, W) - \psi. \tag{5}$$

**Target B (Stochastic/dynamic neighbor intervention).** Let $h^*(\cdot \mid w)$ be a stochastic intervention on $V$ that may depend on $W$. Define

$$\psi \;=\; \Psi(P) \;=\; E_W\Big[ E_{V^* \sim h^*(\cdot \mid W)} \big\{ m(V^*, W) \big\} \Big] \;=\; \iint m(v, w)\, h^*(dv \mid w)\, p_W(dw).$$

The efficient influence function is

$$D^*(O) \;=\; \frac{\bar{h}^*(V \mid W)}{\bar{h}(V \mid W)} \big\{ Y - m(V, W) \big\} \;+\; \int m(v, W)\, h^*(dv \mid W) - \psi. \tag{6}$$

When $V$ is discrete, $\int m(v, W)\, h^*(dv \mid W) = \sum_v m(v, W) h^*(v \mid W)$. Eq. (4) is the node-averaged form of the stochastic-target EIF (6), obtained by averaging $D^*(O_i)$ over units; we write $D^*(O)$ and $\psi$ hereafter.

The multi-view construction only affects the definitions of $V$ and $W$. We now establish the large-sample properties of our estimator. Under Assumptions 1–5, the target parameter is identifiable and the TMLE estimator is consistent and asymptotically normal. In particular, Assumptions 1–3 together with positivity (Assumption 5) ensure identifiability, and Assumption 4 restricts dependence to local neighborhoods, allowing the use of network central limit theorem arguments. As is standard in semiparametric theory, these results hold under mild regularity conditions such as bounded moments and smoothness of the relevant estimators. The detailed conditions we used to obtain the CAN of our estimator are provided in Appendix E.

**Lemma 4.1** (Identification invariance under aggregation). *Let $(W, V)$ be any Borel-measurable aggregations of $(X, T, A^{(1)}, \ldots, A^{(K)})$ with $W$ depending only on pre-treatment inputs. Under Assumptions 1–3 and 5, both targets $\psi_{\text{stat}}, \psi_{\text{stoch}}$ are identified from $P(W, V, Y)$ via*

$$\psi_{\text{stat}} = \int m(v_0, w) \, p_W(dw), \qquad \psi_{\text{stoch}} = \int\int m(v, w) \, h^*(dv \mid w) \, p_W(dw).$$

**Remark 1.** The summary functions $s_X, s_T$ are learned from data via the multi-view GCN modules of Section 5. Lemma 4.1 licenses this: its proof uses only measurability of the aggregation and the pre-treatment property of $W$, so the identifying functionals (Lemma 4.1) and the EIFs in Eqs. (5)–(6) (Proposition E.1 in Appendix E) hold for any admissible aggregation and any number of views $K$, including GNN-learned ones instead of requiring a fixed, pre-specified exposure mapping. In experiments we use $K = 3$.

When the true outcome mechanism operates through an exposure $V^{\text{true}}$ different from the constructed $V$ — that is, $Y_i = f_Y[V_i^{\text{true}}, W_i, \epsilon_{Y_i}]$ with the true interference structure likewise fixed pre-treatment — write $\mu(v, w) = \mathbb{E}[f_Y(v, w, \epsilon_Y)]$ for the true outcome function, $\delta_i = V_i^{\text{true}} - V_i$ for the exposure discrepancy, and $\psi^{\text{true}}$ for the targets of Lemma 4.1 with $\mu$ in place of $m$.

**Proposition 4.1** (Bias bound under approximate exposure sufficiency). *Suppose Assumptions 1–3 and 5 hold for the constructed $(V, W)$, and $\mu(\cdot, w)$ is $L$-Lipschitz uniformly in $w$. Then*

$$\left| \psi_{\text{stat}} - \psi_{\text{stat}}^{\text{true}} \right| \leq L \int \mathbb{E}\big[ |\delta| \,\big|\, V = v_0, \, W = w \big] \, p_W(dw), \qquad \left| \psi_{\text{stoch}} - \psi_{\text{stoch}}^{\text{true}} \right| \leq L \left\| \frac{\bar{h}^*}{\bar{h}} \right\|_\infty \frac{1}{n} \sum_{i=1}^n \mathbb{E} \, |\delta_i|.$$

Assumption 6 is the $\delta \equiv 0$ case, under which $\psi = \psi^{\text{true}}$ exactly; when $\frac{1}{n} \sum_i \mathbb{E}|\delta_i| \to 0$, the guarantees of Theorem 4.1 transfer to $\psi^{\text{true}}$. The proof is given in Appendix E.

**Theorem 4.1** (Consistency and Asymptotic Normality of TMLE). *Let $\hat{\psi}_n$ denote the TMLE for the target parameter $\psi$. Under Assumptions 1–5 and Regularity Conditions (1) and (2),*

$$\sqrt{C_n} \, (\hat{\psi}_n - \psi) \xrightarrow{d} N(0, \sigma^2),$$

*where $C_n = n/(1 + \Delta_n)$ is an effective sample size satisfying $n/K_{\max,n}^2 \lesssim C_n \leq n$, with $\Delta_n \lesssim K_{\max,n}^2$ the maximum degree of the dependency graph (Appendix E). The asymptotic variance $\sigma^2$ equals the variance of the efficient influence function under the true data-generating process.*

## 5   Methodology

In this section, we introduce our *Mvdr* framework, which estimates the causal estimands of interest discussed above. As illustrated in Figure 1, our framework consists of four modules: a network learner that constructs the multi-view networks, confounder representation learning with multi-view feature balancing, a density estimator, and outcome prediction together with the perturbation estimator used for the targeted update. Our motivation is to find a way to integrate information from different views of the network. Inspired by the SEM framework, we propose a novel way to present the transmission of information to units through summary functions and construct them as multi-view networks. Technically, our method offers a semi-parametric way to estimate the two nuisance parameters $h$ and $m$ of the statistical model $\mathcal{M}$. More specifically, the outcome predictor $m$ is a parametric model dependent on the covariate summary $w$ and the exposure summary $v$, and the density estimator $h$ is estimated through every realization of $(w, v)$.

### 5.1 Multi-view Representation of Network

Based on the assumptions of causal relationship, the outcome $Y$ is affected by covariates $X$ through the summary function $W$ and by exposure $T$ through another summary function $V$. In the confounder representation learning module, we use $K$ graph convolutional network (Kipf & Welling, 2017) blocks to represent the summary function $s_{x,i}$ that aggregates information of covariates of unit $i$ and its neighbors on multi-view networks (i.e. $X_i$, $A^{(k)}$, $k = 1, ..., K$):

$$h_i^{(k)} = \sigma\left(\sum_{j \in N_i^{(k)}} \frac{1}{\sqrt{d_i^{(k)} d_j^{(k)}}} W_k^T x_j; \theta_k\right), \tag{7}$$

where $\sigma(\cdot)$ is a sigmoid activation function, $d_i^{(k)}$ is the degree of unit $i$ in the $k$-th network, and $W_k$ is the weight matrix of GCN parameterized by $\theta_k$. To integrate multiple views of structural information and identify the importance of different views of the networks, we use an attention-based fusion module. Given node embeddings $h_i^{(k)} \in \mathbb{R}^d$, $k = 1, \cdots K$ from the GCNs, the scores can be computed as:

$$\alpha_{ik} = \frac{\exp(\mathbf{w}^T h_i^{(k)} + b)}{\sum_{j=1}^K \exp(\mathbf{w}^T h_i^{(j)} + b)}, \tag{8}$$

and the final multi-view representation of unit $i$ can be written as: $w_i = \sum_{k=1}^K \alpha_{ik} \cdot h_i^{(k)}$.

For the exposure summary, we use another stack of GCN layers to learn the aggregation of neighborhood treatment, and then an MLP layer is adopted to learn the weights of exposure from different views. The final exposure mapping $V$ is then combined with two variables: unit treatment $T$ and multi-view neighborhood exposure $S$.

To ensure that the GCNs for different views of the network can learn different aspects of information, we use a ***multi-view network representation discrepancy*** module to maximize the information learned from the multiple GCNs. Specifically, we hope that the summary functions $W_i$, which is used to learn the information from covariates, and $S_i$, which aims to aggregate information from multiple views, can learn different information for the final estimation. In particular, we design a loss function based on the Hilbert-Schmidt independence criterion (HSIC) as follows:

$$L_{HSIC}(W, S) = \frac{1}{n^2} tr(KMLM), \ M = \mathbb{I}_n - \frac{1}{n}\mathbf{1}_n\mathbf{1}_n^T, \tag{9}$$

where $n$ is the number of training units, $\mathbb{I}_n$ represents the identity matrix, and $\mathbf{1}_n$ is the vector of all ones. $K$ and $L$ represent the Gaussian kernel applied to $W$ and $S$, respectively, i.e.,

$$K_{ij} = \exp(-\frac{||w_i - w_j||_2^2}{2}), \ L_{ij} = \exp(-\frac{(s_i - s_j)^2}{2}). \tag{10}$$

### 5.2 Essential Predictors

As we discussed earlier, three essential parts are used to achieve our doubly robust estimator, including the conditional outcome estimator, the conditional density estimator, and the perturbation estimator.

For the ***conditional outcome estimator*** $m$, given the covariate representations $W$ and treatment representations $V$ obtained from the multi-view representation module, we use two multi-layer perceptron (MLP) blocks that consist of three feed-forward layers to infer the outcome $y_i$ for the treated and control groups respectively. This potential outcome predictor is used for initial outcome prediction, representing the outcome regression $m$ of the statistical model $\mathcal{M}$. Specifically, $m_0$ denotes the predictor for the treated units, and $m_1$ denotes the predictor for the controlled units. These potential outcome predictors are optimized by the mean square error (MSE) between the predicted outcomes and observed outcomes:

$$L_1 = \frac{1}{n}\sum_{i=1}^n (m_{t_i}(v_i, w_i) - y_i)^2. \tag{11}$$

The ***conditional density estimator*** $h$ is the conditional distribution of the joint exposure mapping $V = (T, S)$ given covariates $W$, denoted by:

$$h((t, s) \mid w) = \Pr(T = t \mid W = w) \cdot f_{S|T,W}(s \mid t, w). \tag{12}$$

This factorization separates the discrete ego treatment $T$ from the continuous neighbor summary $S$.

*Ego Treatment Model.* We estimate $\Pr(T = 1 \mid W = w)$ using an MLP classifier. Given input $w$, the network outputs $\hat{e}(w) = \sigma(g_\theta(w))$, where $g_\theta$ is an MLP and $\sigma$ is the sigmoid function. The model is trained by minimizing the binary cross-entropy loss:

$$L_e(\theta) = -\frac{1}{n} \sum_{i=1}^{n} \Big[ T_i \log \hat{e}(W_i) + (1 - T_i) \log(1 - \hat{e}(W_i)) \Big]. \tag{13}$$

*Neighbor Summary Model.* For the second factor $f_{S|T,W}$, we approximate the conditional density of $S$ given $(T, W)$. Following the conditional histogram approach (Díaz Muñoz & Van Der Laan, 2011), we normalize $v$ into $[0, 1]$ and partition its support into $C$ intervals. For each interval, we fit a multinomial logistic regression model that outputs bin-wise probabilities:

$$\pi_i = \text{softmax}\big(W_{\text{dens}}^\top(x_i \| w_i) + b\big) \in \mathbb{R}^C, \tag{14}$$

where $W_{\text{dens}}$ are learnable parameters and $\|$ denotes concatenation. The conditional density at $v_i$ is then obtained by interpolating between the corresponding lower and upper bins:

$$\hat{h}(v_i \mid x_i, w_i) = \pi_{i,L_i} + (v_i \cdot C - L_i) \cdot \big(\pi_{i,U_i} - \pi_{i,L_i}\big), \tag{15}$$

where $L_i$ and $U_i$ denote the indices of the bins bracketing $v_i$. This estimator can be viewed as a flexible, data-adaptive approximation to the Radon–Nikodým derivative of the target intervention distribution with respect to the observed distribution. It is trained by minimizing the negative log-likelihood:

$$L_n = -\sum_{i=1}^{n} \log \hat{h}(v_i \mid x_i, w_i). \tag{16}$$

Putting all the two density models together, the final density model is trained by minimizing the negative log-likelihood:

$$L_2 = L_e + L_n = -\sum_{i=1}^{n} \log \hat{h}((x_i, s_i) \mid w_i). \tag{17}$$

To achieve doubly robust estimation, we use an MLP block to learn $\epsilon$ that makes the influence function equal to 0. [1] The loss function of our **perturbation estimator** is:

$$L_3 = \frac{1}{n} \sum_{i=1}^{n} \big(y_i - \hat{m}_i - \epsilon H_i\big)^2, \qquad H_i = \frac{\hat{h}^*(v_i, w_i)}{\hat{h}(v_i, w_i)}. \tag{18}$$

Combining the three modules for our estimation together, we have the final loss function:

$$Loss = \sum_{i=1}^{n}(y_i - \widetilde{y}_i)^2 - \sum_{i=1}^{n} \log \hat{h}((x_i, s_i) \mid w_i) + \sum_{i=1}^{n}(y_i - \hat{m}_i - \epsilon H_i)^2 + \lambda L_{HSIC}. \tag{19}$$

## 5.3 Inference under Interventions Policies

After training the initial outcome predictor $m$ and the density estimator $h$, we propose the procedure of predicting average potential outcome under certain intervention policy. Suppose our intervention towards the network is $T^* \in \{0, 1\}^n$. We first calculate the initial potential outcome $\hat{y}_i = m_{t_i^*}(v_i^*, w_i)$. Secondly, to achieve the robust estimator of the final result, we need to compute the auxiliary weights as:

$$H_i = \frac{\hat{h}^*(v_i^*, w_i)}{\hat{h}(v_i, w_i)} \tag{20}$$

---

[1] Standard TMLE for finite-dimensional parameters fits a scalar $\epsilon$. In our setting, the exposure summary $V = (T, S)$ contains a continuous component $S$, so the clever covariate varies with $v$. Following Nie et al. (2021) and the continuous-treatment TMLE literature, we parameterize $\epsilon(v)$ as a B-spline network to flexibly cover the continuous support of $V$. When $V$ is discrete (static intervention, Target A), $\epsilon$ reduces to a finite-dimensional vector, consistent with the standard formulation.

For the last step, we construct the final outcome prediction with targeted learning. The final estimation for the unit $i$ can be written as:

$$\widetilde{Y}_i^* = m_{t_i^*}(v_i^*, w_i) + \epsilon(v_i^*, w_i) \cdot H_i, \tag{21}$$

and the average potential outcome under the intervention $T^*$ is given by:

$$\bar{\widetilde{Y}}^* = \frac{1}{n} \sum_{i=1}^n \widetilde{Y}_i^*. \tag{22}$$

This estimator is doubly robust: it is consistent for the target parameter $\psi$ if either $h$ or $m$ is correctly specified.

## 6 Experiment

In this section, we validate the proposed method *Mvdr* on two commonly used semi-synthetic datasets. We also construct a new semi-synthetic dataset based on the DBLP dataset. We verify the effectiveness of our method and further evaluate the correctness of our analysis using these semi-synthetic datasets. In particular, we aim to answer the following research questions (RQ):

- RQ1: How does the proposed method compare with existing methods in terms of effect estimation performance?

- RQ2: How does the perturbation estimator module affect the performance of our methods?

- RQ3: Does our method stably perform well under different types of misspecification?

In the following, we first introduce the experimental setup and then answer the questions above by conducting the corresponding experiments.

**Datasets.** In our experiment, we utilize two widely used semi-synthetic datasets, i.e., *BlogCatalog* and *Flickr*. Another contribution of this paper is that we construct a new semi-synthetic dataset based on the *DBLP* dataset. The DBLP dataset contains rich information of conference and journal publications, such as detailed information of authors. We construct a coauthorship network by matching the author information and the publication information. Detailed statistics of three datasets are provided in Appendix C.

**Multi-view network construction.** A simple idea of constructing multi-view networks is to compute the similarity scores of covariates between units and construct the relational networks based on their similarity. We follow the multi-view network construction method from (Lin et al., 2023), in which multi-view networks were constructed for the BlogCatalog and Flickr datasets based on the similarity of covariates. The constructed latent networks can then be potential for the upcoming multi-view information aggregation. These latent multi-view networks are learned from the covariates and are not directly observed from the data.

Since it is impossible to observe the counterfactual outcomes, we need to mimic the unknown output. Following prior studies on ITE and ATE, we simulate outcomes as ground-truth values for counterfactual outcomes that are not available using the following equation:

$$Y_i = \beta_T \cdot T_i + \beta_X \cdot \left(\mathbf{w}^\top \mathbf{x}_i\right) + \beta_{NX} \cdot \sum_{k=1}^3 \frac{1}{|\mathcal{N}_i^{(k)}|} \sum_{j \in \mathcal{N}_i^{(k)}} \mathbf{w}^\top \mathbf{x}_j + \beta_{NT} \cdot \left(\sum_{k=1}^3 \alpha_k \cdot \frac{\sum_{j \in \mathcal{N}_i^{(k)}} T_j}{|\mathcal{N}_i^{(k)}|}\right) + \varepsilon_i. \tag{23}$$

This construction method assumes that the exposure mapping is a simple function of the ratio of treated neighbors across views. Here, $\mathbf{x}_i \in \mathbb{R}^d$ denotes the covariate vector for unit $i$, and $T_i \in \mathbb{R}$ is the assigned treatment. The neighborhood sets $\mathcal{N}_i^{(k)}$ are derived from three adjacency matrices $A^{(1)}, A^{(2)}, A^{(3)}$, representing multi-view network structures. Each network layer $k \in \{1, 2, 3\}$ contributes to the potential outcome through aggregated covariate signals and treatment exposures, weighted by coefficients $\alpha_k$. We set $\alpha_1 = \alpha_2 = 1$ and $\alpha_3 = 0.5$ in our experiments to reflect the varying network influence. Lastly, $\varepsilon_i \sim \mathcal{N}(0, 1)$ introduces stochastic variation.

Table 1: Comparison of methods on causal effect of network estimand with different interventions. 'Static0' sets all units to control; 'Static1' sets all to treated; 'Stochastic' applies treatment to 35% randomly; 'Dynamic' assigns treatment based on covariate-dependent probabilities. Best performance in each row is **bolded**. Results are reported as the mean and standard deviation over five runs for each setting.

| Dataset | | Intervention | CFR | CFR+z | ND | ND+z | TARNet | TARNet+z | NetEst | HINITE | TNet | Mvdr(w/o. $\mathcal{L}_3$) | Mvdr |
|---|---|---|---|---|---|---|---|---|---|---|---|---|---|
| BlogCata | With-in-Sample | Static0 | $0.4428_{0.2618}$ | $\mathbf{0.4034}_{0.2625}$ | $0.4466_{0.2508}$ | $0.4093_{0.2576}$ | $0.4457_{0.2442}$ | $0.4082_{0.2726}$ | $0.3861_{0.2576}$ | $1.0292_{0.7670}$ | $16.346_{17.3932}$ | $0.4968_{0.2366}$ | $0.4891_{0.1899}$ |
| | | Static1 | $0.5381_{0.1829}$ | $0.4302_{0.2203}$ | $0.5766_{0.1959}$ | $0.3953_{0.2357}$ | $0.5873_{0.2077}$ | $0.5045_{0.1933}$ | $0.8412_{0.2356}$ | $1.2021_{0.9884}$ | $7999.1908_{12974.5585}$ | $0.3035_{0.1633}$ | $\mathbf{0.2574}_{0.1515}$ |
| | | Stochastic | $0.5833_{0.2865}$ | $0.5747_{0.2845}$ | $0.5821_{0.3006}$ | $0.5797_{0.2920}$ | $0.5881_{0.3175}$ | $0.5813_{0.3119}$ | $0.2751_{0.1782}$ | $0.6227_{0.2099}$ | $81.2073_{123.1866}$ | $0.2818_{0.1699}$ | $\mathbf{0.2668}_{0.1847}$ |
| | | Dynamic | $0.4858_{0.2952}$ | $0.4869_{0.3001}$ | $0.4910_{0.3025}$ | $0.4974_{0.3033}$ | $0.4919_{0.3108}$ | $0.5065_{0.3007}$ | $0.5001_{0.3032}$ | $0.5585_{0.2657}$ | $23.1303_{47.0434}$ | $0.4486_{0.3433}$ | $\mathbf{0.4401}_{0.3636}$ |
| | Out-of-Sample | Static0 | $0.4251_{0.2345}$ | $0.3723_{0.2388}$ | $0.4316_{0.1992}$ | $\mathbf{0.3283}_{0.2780}$ | $0.4758_{0.2521}$ | $0.5224_{0.2921}$ | $0.4005_{0.348}$ | $0.9796_{0.7588}$ | $15.6192_{14.4018}$ | $0.4509_{0.2701}$ | $0.4174_{0.3391}$ |
| | | Static1 | $0.5011_{0.1543}$ | $0.3958_{0.2316}$ | $0.4179_{0.4720}$ | $0.5031_{0.3988}$ | $0.4843_{0.3408}$ | $0.3502_{0.3040}$ | $0.7291_{0.3454}$ | $1.1732_{0.9409}$ | $5735.9332_{9219.6814}$ | $0.2940_{0.2183}$ | $\mathbf{0.2279}_{0.1549}$ |
| | | Stochastic | $0.4755_{0.2916}$ | $0.4590_{0.2745}$ | $0.4478_{0.3245}$ | $0.4914_{0.3420}$ | $0.5313_{0.3889}$ | $0.5067_{0.4134}$ | $0.4446_{0.3048}$ | $0.5065_{0.3123}$ | $112.701_{133.1668}$ | $0.1974_{0.1821}$ | $\mathbf{0.1847}_{0.1784}$ |
| | | Dynamic | $0.6421_{0.3101}$ | $0.6385_{0.3204}$ | $0.6324_{0.3334}$ | $0.5830_{0.3148}$ | $0.6721_{0.3047}$ | $0.6414_{0.3250}$ | $0.6623_{0.3347}$ | $0.7148_{0.3062}$ | $18.7178_{39.9793}$ | $0.4785_{0.3798}$ | $\mathbf{0.4500}_{0.3886}$ |
| Flickr | With-in-Sample | Static0 | $0.1710_{0.0766}$ | $\mathbf{0.0887}_{0.0986}$ | $0.1747_{0.0768}$ | $0.1002_{0.0231}$ | $0.1816_{0.0700}$ | $0.1700_{0.0676}$ | $0.1014_{0.0747}$ | $0.3486_{0.3168}$ | $5.9266_{5.5908}$ | $0.1311_{0.0371}$ | $0.1086_{0.0896}$ |
| | | Static1 | $0.1751_{0.1185}$ | $\mathbf{0.0994}_{0.0327}$ | $0.2522_{0.1844}$ | $0.0923_{0.0613}$ | $0.2683_{0.1981}$ | $0.1347_{0.0902}$ | $0.4473_{0.0706}$ | $1.1022_{0.8934}$ | $18.1979_{16.6327}$ | $0.3878_{0.0494}$ | $0.3788_{0.0658}$ |
| | | Stochastic | $0.1410_{0.0777}$ | $0.1368_{0.0967}$ | $0.1647_{0.0974}$ | $0.1908_{0.1021}$ | $0.1779_{0.1059}$ | $0.2181_{0.0993}$ | $0.3243_{0.0986}$ | $0.2045_{0.1504}$ | $10.8568_{11.5352}$ | $0.1995_{0.0742}$ | $\mathbf{0.1316}_{0.0375}$ |
| | | Dynamic | $0.1793_{0.1497}$ | $0.2085_{0.0617}$ | $0.2109_{0.1635}$ | $0.2305_{0.0678}$ | $0.2279_{0.1714}$ | $0.1905_{0.0503}$ | $0.1436_{0.1705}$ | $0.5980_{0.1442}$ | $2934.7736_{3022.9666}$ | $0.0401_{0.041}$ | $\mathbf{0.0338}_{0.0325}$ |
| | Out-of-Sample | Static0 | $\mathbf{0.1241}_{0.0995}$ | $0.1407_{0.1073}$ | $0.1419_{0.0812}$ | $0.1391_{0.1172}$ | $0.1642_{0.0738}$ | $0.1289_{0.1133}$ | $0.1376_{0.0892}$ | $0.3111_{0.2683}$ | $14.9494_{19.1093}$ | $0.2731_{0.0744}$ | $0.2163_{0.0486}$ |
| | | Static1 | $0.1927_{0.1098}$ | $\mathbf{0.1565}_{0.1069}$ | $0.2960_{0.1432}$ | $0.2018_{0.0665}$ | $0.3354_{0.1231}$ | $0.2048_{0.0773}$ | $0.2352_{0.0867}$ | $1.0141_{0.8732}$ | $26.4602_{21.5615}$ | $0.634_{0.1683}$ | $0.4648_{0.1063}$ |
| | | Stochastic | $0.1424_{0.1079}$ | $0.1598_{0.0834}$ | $0.2542_{0.1024}$ | $0.1864_{0.0732}$ | $0.1471_{0.1169}$ | $0.2180_{0.1140}$ | $0.3173_{0.0793}$ | $0.2248_{0.1756}$ | $4.875_{5.5107}$ | $0.1245_{0.0409}$ | $\mathbf{0.1172}_{0.0718}$ |
| | | Dynamic | $0.1642_{0.1354}$ | $0.1703_{0.1496}$ | $0.2032_{0.1661}$ | $0.2905_{0.1676}$ | $0.2036_{0.1606}$ | $0.1293_{0.0574}$ | $0.7368_{0.1807}$ | $0.3059_{0.2397}$ | $2141.9216_{1912.0244}$ | $0.0470_{0.0152}$ | $\mathbf{0.0402}_{0.0287}$ |
| DBLP | With-in-Sample | Static | $0.6801_{0.6428}$ | $0.7406_{0.6214}$ | $0.8354_{0.5585}$ | $0.6755_{0.5753}$ | $0.9776_{0.8482}$ | $0.7969_{0.6272}$ | $0.9383_{0.593}$ | $0.8162_{0.5422}$ | $7.2816_{17.7863}$ | $0.7142_{0.4798}$ | $\mathbf{0.5933}_{0.5328}$ |
| | | Static1 | $0.6128_{0.4203}$ | $0.6682_{0.4753}$ | $0.6194_{0.3356}$ | $0.7100_{0.6311}$ | $\mathbf{0.5597}_{0.3182}$ | $0.6378_{0.4699}$ | $0.697_{0.2554}$ | $1.4879_{1.1256}$ | $25.8228_{50.5348}$ | $1.1797_{0.5966}$ | $1.1800_{0.5476}$ |
| | | Stochastic | $0.6548_{0.5236}$ | $0.7344_{0.5142}$ | $0.7434_{0.4676}$ | $0.8192_{0.5609}$ | $0.7264_{0.4199}$ | $0.8145_{0.5024}$ | $0.6705_{0.427}$ | $1.0153_{0.6379}$ | $28.2031_{124.7104}$ | $0.8348_{0.5777}$ | $\mathbf{0.6602}_{0.4815}$ |
| | | Dynamic | $0.5924_{0.5628}$ | $0.5944_{0.5645}$ | $0.5956_{0.4799}$ | $0.5701_{0.6118}$ | $0.7413_{0.6965}$ | $0.7391_{0.7125}$ | $0.6482_{0.6376}$ | $0.7816_{0.7768}$ | $7.0901_{25.3574}$ | $0.7038_{0.5519}$ | $\mathbf{0.5633}_{0.5301}$ |
| | Out-of-Sample | Static | $0.6520_{0.6234}$ | $0.7282_{0.5922}$ | $0.8258_{0.5324}$ | $0.756_{0.5221}$ | $0.8693_{0.7477}$ | $0.8637_{0.5882}$ | $0.976_{0.5703}$ | $1.0222_{0.4386}$ | $5.033_{16.6037}$ | $0.6208_{0.4500}$ | $\mathbf{0.5369}_{0.5328}$ |
| | | Static1 | $0.6532_{0.4305}$ | $0.7056_{0.4685}$ | $0.6401_{0.4714}$ | $0.6800_{0.4572}$ | $0.6391_{0.4214}$ | $\mathbf{0.5417}_{0.3613}$ | $0.6841_{0.3559}$ | $1.3416_{1.3356}$ | $11.0473_{13.9739}$ | $0.7730_{0.5636}$ | $0.7018_{0.5807}$ |
| | | Stochastic | $0.8120_{0.5061}$ | $0.7946_{0.4904}$ | $0.7804_{0.4153}$ | $0.7830_{0.5038}$ | $0.7610_{0.3402}$ | $0.7142_{0.4665}$ | $0.976_{0.5703}$ | $0.7249_{0.6312}$ | $26.2156_{117.6624}$ | $0.7861_{0.5641}$ | $\mathbf{0.6711}_{0.4551}$ |
| | | Dynamic | $0.5776_{0.6066}$ | $0.5801_{0.6092}$ | $0.5981_{0.5365}$ | $0.6501_{0.6210}$ | $0.6001_{0.7037}$ | $0.7048_{0.7051}$ | $0.7342_{0.672}$ | $0.8196_{0.6017}$ | $1.7650_{2.5869}$ | $0.6913_{0.5283}$ | $\mathbf{0.5628}_{0.4488}$ |

## 6.1 Three Types of Interventions

To evaluate the stability of our method and baselines under different types of interventions, we design three interventions to mimic different hypothetical treatment policies in real life. **Static intervention** assigns all the units in the network as treated or not treated. **Dynamic intervention** assigns exposures as a user-specified, deterministic function of covariates. This scenario often appears when there are some constraints in resources. For example, the economic incentive was resource constrained and could only be allocated to up to 10% of the whole group. In our experiments, we allocate the top 10% most connected members of the community. **Stochastic intervention** assigns exposures as a user-specified, random function. In our experiments, we assign each unit to treatment with a constant probability of 0.35.

## 6.2 Baselines and Metrics

**Baselines.** We compare our method with several representative baselines in causal inference, including methods that deal with a single observed network and multi-view networks. CFR (Johansson et al., 2023) and ND (Guo et al., 2019) are modified with additionally inputting the exposure. TARNet (Shalit et al., 2017) has a similar model architecture as CFR but removes the balance term. For fair comparison, we augment CFR, ND, and TARNet with the exposure summary $z_i = \frac{1}{|\mathcal{N}_i|} \sum_{j \in \mathcal{N}_i} T_j$ as an additional input feature, denoted CFR+z, ND+z, and TARNet+z respectively. HINITE (Lin et al., 2023) considers several observed heterogeneous networks. NetEst (Jiang & Sun, 2022) is designed for treatment effect estimation under interference. TNet (Chen et al., 2024) uses the proportion of treated neighbors as the exposure mapping, which is a doubly robust method.

**Evaluation Metric.** The main purpose of our model is to predict the average expected outcome of hypothetical intervention policies. We evaluate the accuracy of the model predicting the average expected outcome of a group receiving treatment under different intervention policies, written as:

$$\varepsilon_{intervention} = |\widetilde{Y}_n^* - Y_n^*|, \text{ where } Y_n^* = \frac{1}{n} \sum_{i=1}^{n} Y_i^*.$$

## 6.3 Results and Ablation Studies

**Comparisons with Baselines.** Results in Table 1 show that our Mvdr method outperforms baselines in most cases, especially for the stochastic and dynamic policy settings, where the treatment assignment is related to units' covariates and neighborhood exposure. Under static interventions, strong i.i.d. baselines can be competitive or best, suggesting that static policies are less sensitive to misspecification of the exposure mechanism. We also report the results of some other metrics in Appendix F.

Table 2: Stochastic Policy Evaluation. The proportion of treated units ranges from 0.20 to 0.80, reflecting real-world policy settings where the treatment size may fluctuate according to budget constraints. Best performance in each row is **bolded**.

| Dataset | Proportion | CFR | CFR+z | ND | ND+z | TARNet | TARNet+z | NetEst | HINITE | TNet | Mvdr |
|---|---|---|---|---|---|---|---|---|---|---|---|
| BlogCata | 0.20 | $0.3197_{0.305}$ | $0.3562_{0.3169}$ | $0.5202_{0.4264}$ | $0.5617_{0.4721}$ | $0.4834_{0.364}$ | $0.4491_{0.3787}$ | $0.4672_{0.4064}$ | $0.6987_{0.2483}$ | $57.2356_{57.4217}$ | $\mathbf{0.3016}_{0.2646}$ |
| | 0.50 | $0.2598_{0.2665}$ | $0.2552_{0.261}$ | $0.3095_{0.2291}$ | $0.3846_{0.2485}$ | $0.3743_{0.3218}$ | $0.2980_{0.3026}$ | $0.313_{0.4055}$ | $0.5063_{0.2674}$ | $8.5232_{11.3317}$ | $\mathbf{0.2948}_{0.2602}$ |
| | 0.65 | $0.4678_{0.3421}$ | $0.494_{0.3619}$ | $0.4957_{0.2699}$ | $0.6111_{0.3331}$ | $0.4117_{0.2646}$ | $0.4747_{0.2324}$ | $0.5033_{0.3818}$ | $1.0180_{0.5412}$ | $7.3843_{6.2696}$ | $\mathbf{0.3990}_{0.3953}$ |
| | 0.80 | $0.2743_{0.175}$ | $0.2728_{0.1868}$ | $0.2859_{0.2169}$ | $0.3844_{0.2375}$ | $0.2581_{0.1462}$ | $0.3422_{0.1296}$ | $0.4369_{0.3256}$ | $0.9691_{0.6849}$ | $3033.8552_{2888.6561}$ | $\mathbf{0.2300}_{0.1513}$ |
| Flickr | 0.20 | $0.0956_{0.0763}$ | $0.1181_{0.0968}$ | $0.1745_{0.0364}$ | $0.1864_{0.0732}$ | $0.0992_{0.0541}$ | $0.1294_{0.055}$ | $0.1724_{0.0824}$ | $0.4369_{0.3551}$ | $320.0_{634.0}$ | $\mathbf{0.0872}_{0.0694}$ |
| | 0.50 | $0.1150_{0.0419}$ | $0.1161_{0.0507}$ | $0.0966_{0.0486}$ | $0.1277_{0.0537}$ | $0.1414_{0.0649}$ | $0.1293_{0.0574}$ | $0.2080_{0.1165}$ | $0.1928_{0.2158}$ | $320.5_{545.6}$ | $\mathbf{0.0914}_{0.0688}$ |
| | 0.65 | $0.1028_{0.0692}$ | $0.0966_{0.0706}$ | $0.1118_{0.0111}$ | $0.1002_{0.0527}$ | $0.1071_{0.0273}$ | $0.1036_{0.026}$ | $0.2436_{0.0739}$ | $0.4003_{0.3475}$ | $333.2976_{295.757}$ | $\mathbf{0.0908}_{0.0614}$ |
| | 0.80 | $0.1196_{0.0886}$ | $0.0933_{0.0495}$ | $0.1781_{0.0703}$ | $0.137_{0.0744}$ | $0.2265_{0.110}$ | $0.2008_{0.0558}$ | $0.5136_{0.1013}$ | $0.5119_{0.3149}$ | $1073.9346_{1198.002}$ | $\mathbf{0.0850}_{0.0700}$ |
| DBLP | 0.20 | $0.6884_{0.6434}$ | $0.4900_{0.6684}$ | $0.5301_{0.6866}$ | $0.4749_{0.7014}$ | $0.5244_{0.6407}$ | $0.5582_{0.7820}$ | $0.6222_{0.7431}$ | $0.8006_{0.7482}$ | $121.7685_{240.3945}$ | $\mathbf{0.4284}_{0.5858}$ |
| | 0.50 | $0.4794_{0.3694}$ | $0.4749_{0.3688}$ | $0.7325_{0.5254}$ | $0.5381_{0.4129}$ | $0.5989_{0.4951}$ | $0.5228_{0.3430}$ | $0.5057_{0.3798}$ | $0.8234_{0.6519}$ | $3.0800_{4.7400}$ | $\mathbf{0.4363}_{0.3654}$ |
| | 0.65 | $0.5841_{0.5569}$ | $0.6119_{0.5669}$ | $0.5515_{0.2723}$ | $0.5838_{0.6110}$ | $0.5397_{0.2044}$ | $0.5593_{0.4905}$ | $0.5499_{0.4583}$ | $1.4967_{1.0754}$ | $17.701_{32.4764}$ | $\mathbf{0.5140}_{0.4782}$ |
| | 0.80 | $0.6107_{0.6391}$ | $0.6028_{0.6532}$ | $0.7433_{0.2997}$ | $0.5800_{0.6448}$ | $0.7180_{0.2523}$ | $0.6690_{0.5468}$ | $0.6785_{0.3982}$ | $1.7752_{1.2415}$ | $9.3042_{7.7427}$ | $\mathbf{0.4026}_{0.5439}$ |

**The Perturbation Estimator.** As shown in Table 1, we have conducted experiments to compare the performance between Mvdr and Mvdr(w/o. $\mathcal{L}_3$) across the three datasets. Overall, Mvdr outperforms Mvdr(w/o. $\mathcal{L}_3$), which is as expected. Results show that the TMLE framework introduces double robustness to Mvdr and mitigates bias through the three modules. When the initial outcome model includes bias, the conditional density works as a de-biasing step to produce robust results.

**The Stochastic Policy.** We evaluate the performance when the treated proportion of stochastic assignment policy changes from 0.20 to 0.80. As shown in Table 2, Mvdr outperforms baselines across different assignment policy settings.

**Unbiasedness under Misspecification.** To show that our model performs stably when either working model $m$ or $h$ is misspecified, we simulate data mimicking two misspecification regimes. (1) Flipping recorded treatments at rate $\rho \in [0.25, 1]$ corrupts the constructed exposure while leaving the outcome-generating exposure unchanged, increasing the exposure discrepancy $\mathbb{E}|\delta|$ of Proposition 4.1 in proportion to $\rho$. (2) Introducing a tanh nonlinearity in the outcome function violates exact exposure sufficiency (Assumption 6) while preserving the Lipschitz condition of Proposition 4.1 and leaving the treatment mechanism, hence $h$, correctly specified. Figure 2 and Figure 3 show that the error under misspecification grows slowly and remains close to the original results, matching the linear degradation predicted by Proposition 4.1.

## 7 Discussions and Conclusions

This paper proposes a framework to estimate policy values under interventions on constructed multi-view exposures, over a heterogeneous network with latent dependency, through a targeted maximum likelihood estimation method. We provided theorems showing the consistency and double robustness of the estimator, and a bias bound relating our estimand to the effect on the true interference structure. Our estimand is an implementable policy value that is identified and interpretable regardless of the missingness in the observed network; its coincidence with the true-network effect is governed by the exposure discrepancy $\mathbb{E}|\delta|$ (Proposition 4.1), which becomes partially estimable given validation data on true edges (Chao et al., 2025) — a direction we leave to future work. Experiments show that our method performs well on estimating average expected outcomes under hypothetical intervention policies on networks. Our method aims to solve the problem of latent network dependency in causal inference under heterogeneous network setting, by modeling potential outcome as conditional expectation of two network related variable, $W$ and $V$. Though it achieves relatively good performance, we acknowledge that this method has its limitations. For example, the neighbourhood exposure summary $S$ is one-dimensional, which may not suffice to capture the interference mechanism in complex network settings. We plan to explore other semi-parametric methods to model the exposure mapping and achieve better identification of interference in our future work.

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

## A  Notations

| Symbol | Type / domain | Description |
|---|---|---|
| *Units, data, and network* | | |
| $i \in \{1, \ldots, n\}$ | index | Unit (node) index; $n$ is network size. |
| $x_i$ | $\mathbb{R}^d$ | Covariates of unit $i$ (column vector). |
| $t_i$ | $\{0,1\}$ | Binary treatment of unit $i$. |
| $y_i$ | $\mathbb{R}$ | Observed outcome for unit $i$ under $t_i$. |
| $X$ | $\mathbb{R}^{n \times d}$ | Covariate matrix $X = (x_1, \ldots, x_n)^{\top}$. |
| $T$ | $\{0,1\}^n$ | Treatment vector $T = (t_1, \ldots, t_n)^{\top}$. |
| $Y$ | $\mathbb{R}^n$ | Outcome vector $Y = (y_1, \ldots, y_n)^{\top}$. |
| $A^{(k)}$ | $\{0,1\}^{n \times n}$ | Adjacency of the $k$-th pre-treatment view (directly observed or constructed from covariates), $k = 1, \ldots, K$; symmetric. |
| $K$ | $\mathbb{N}$ | Number of views. |
| $\tilde{A}$ | $\{0,1\}^{n \times n}$ | Union support graph: $\tilde{A}_{ij} = 1$ iff $A_{ij}^{(k)} = 1$ for at least one view $k$. Used in Eq. (1) and Assumption 4. |
| $N_i^{(k)}$ | subset of $\{1, \ldots, n\}$ | Neighbor set of node $i$ in view $k$. |
| $\tilde{N}_i$ | subset of $\{1, \ldots, n\}$ | Neighbor set of node $i$ in $\tilde{A}$. |
| $d_i^{(k)}$ | $\mathbb{N}$ | Degree of node $i$ in view $k$ (in $A^{(k)}$). |
| *Structural equation model (SEM)* | | |
| $f_X, f_T, f_Y$ | functions | Unknown structural functions in the working SEM, Eq. (1). |
| $\varepsilon_{X_i}, \varepsilon_{T_i}, \varepsilon_{Y_i}$ | r.v. | Exogenous errors for unit $i$. |
| $s_X^{(k)}, s_T^{(k)}$ | summary maps | View-specific summaries of $\{X_j : A_{ij}^{(k)} = 1\}$ and $\{T_j : A_{ij}^{(k)} = 1\}$. |
| $\mathrm{Agg}_W, \mathrm{Agg}_V$ | aggregators | Borel-measurable maps fusing view-specific summaries. |
| $W_i$ | r.v. | Aggregated covariate summary $W_i = \mathrm{Agg}_W(W_i^{(1)}, \ldots, W_i^{(K)})$. |
| $S_i$ | r.v. | Aggregated neighbor exposure summary of treatments. |
| $V_i$ | r.v. | Joint exposure summary, typically $V_i = (T_i, S_i)$. |
| *Statistical models and conditionals* | | |
| $m(v, w)$ | $\mathbb{R}$ | Outcome regression: $m(v,w) = \mathbb{E}[Y \mid V = v, W = w]$. |
| $h(v \mid w)$ | density / pmf | Exposure mechanism: $h(v \mid w) = \mathbb{P}(V = v \mid W = w)$. |
| $h(v, w)$ | joint | Joint law of $(V, W)$; $h(v,w) = \mathbb{P}(V = v, W = w)$. |
| $h^*(\cdot \mid w)$ | intervention kernel | Target (stochastic) intervention on $V$ given $W = w$. |
| $p_W$ | law on $\mathcal{W}$ | Marginal distribution of $W$. |
| $O$ | tuple | Observed variables $O = (W, V, Y)$. |
| $\bar{h}(\cdot)$ | average | Node-wise average: e.g. $\bar{h}(v, w) = \frac{1}{n} \sum_{j=1}^n h_j(v, w)$. |
| $\bar{h}^*(\cdot)$ | average under $h^*$ | $\bar{h}^*(v, w) = \frac{1}{n} \sum_{j=1}^n h_j^*(v, w)$. |
| *Targets / estimands* | | |
| $T^*$ | $\{0,1\}^n$ | Hypothetical intervention assignment over the network. |
| $V^*$ | r.v. | Exposure under intervention, e.g. $V^* \sim h^*(\cdot \mid W)$. |
| $v_0$ | value in $\mathsf{supp}(V)$ | Fixed (static) neighbor exposure level. |
| $\psi$ | $\mathbb{R}$ | Target parameter, defined on the constructed exposures (Section 3.4): $\mathbb{E}[m(v_0, W)]$ or $\mathbb{E}_W \int m(v, W) h^*(dv \mid W)$. |
| $H_i$ | weight | "Clever covariate": $\frac{\bar{h}^*(V_i \mid W_i)}{\bar{h}(V_i \mid W_i)}$ (stochastic) or $\frac{\mathbf{1}\{V_i = v_0\}}{\bar{h}(v_0 \mid W_i)}$ (static). |
| $D^*(O)$ | r.v. | Efficient influence function (EIF) for $\psi$. |
| *Network misspecification (Proposition 4.1)* | | |
| $V_i^{\mathrm{true}}$ | r.v. | Exposure induced by the true (unobserved) interference structure. |
| $\mu(v, w)$ | $\mathbb{R}$ | True outcome function: $\mu(v, w) = \mathbb{E}[f_Y(v, w, \varepsilon_Y)]$. |
| $\delta_i$ | r.v. | Exposure discrepancy $\delta_i = V_i^{\mathrm{true}} - V_i$; unobserved. |
| $\psi^{\mathrm{true}}$ | $\mathbb{R}$ | Target defined on the true exposures; $|\psi - \psi^{\mathrm{true}}| \leq L B \frac{1}{n} \sum_i \mathbb{E}|\delta_i|$. |
| $L$ | constant | Lipschitz constant of $\mu(\cdot, w)$, uniform in $w$. |
| $B$ | constant | Bound on the density ratio $h^*/h$; equals $\|\bar{h}^*/\bar{h}\|_\infty$ in Proposition 4.1. |
| *Asymptotics / dependence* | | |
| $K_{\max,n}$ | $\mathbb{N}$ | Maximum degree of the union support graph $\tilde{A}$. |
| $\Delta_n$ | $\mathbb{N}$ | Maximum degree of the dependency graph of $\{D^*(O_i)\}$; $\Delta_n \lesssim K_{\max,n}^2$. |
| $C_n$ | $\mathbb{N}$ | Effective sample size, $C_n = n/(1 + \Delta_n)$; satisfies $n/K_{\max,n}^2 \lesssim C_n \leq n$. |

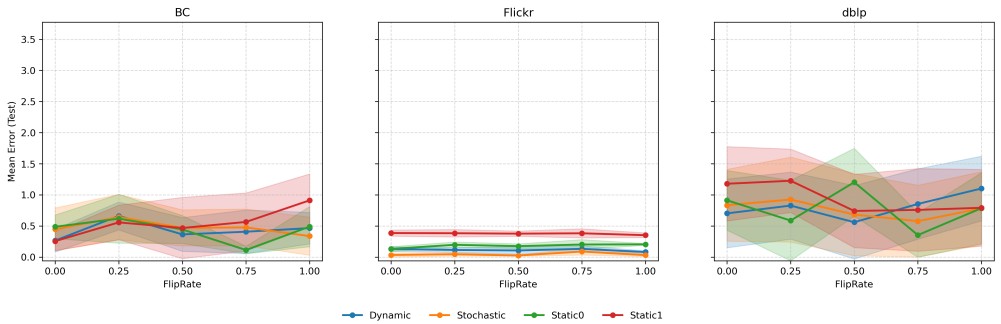

Figure 2: Evaluation on bias caused by misspecification on treatment mechanism.

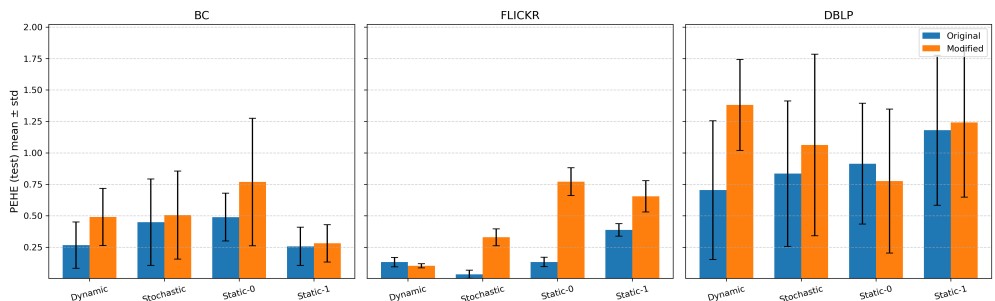

Figure 3: Evaluation on bias caused by misspecification on outcome model.

## B  Figures

## C  Data Statistics

| Dataset | Graph | Nodes | Edges | AvgDeg | StdDeg | MinDeg | MaxDeg | Density | Components |
|---------|-------|-------|-------|--------|--------|--------|--------|---------|------------|
| BC | A1 | 5196 | 171743 | 66 | 54.83 | 5 | 769 | 0.012725 | 1 |
| | A2 | 5196 | 32036 | 12 | 1.77 | 10 | 20 | 0.002374 | 1 |
| | A3 | 5196 | 34957 | 13 | 2.12 | 10 | 23 | 0.002590 | 1 |
| Flickr | A1 | 7301 | 238086 | 65 | 132.66 | 2 | 1834 | 0.008934 | 1 |
| | A2 | 7301 | 45727 | 12 | 2.07 | 10 | 30 | 0.001716 | 1 |
| | A3 | 7301 | 48203 | 13 | 2.18 | 10 | 32 | 0.001809 | 1 |
| DBLP | A1 | 2828 | 14638 | 10 | 14.47 | 1 | 262 | 0.003662 | 1 |
| | A2 | 2828 | 22718 | 16 | 11.78 | 10 | 200 | 0.005683 | 2 |
| | A3 | 2828 | 22291 | 15 | 8.18 | 10 | 86 | 0.005576 | 1 |

## D  DBLP Dataset Construction

We construct a new semi-synthetic dataset from the AMiner academic corpus (Tang et al., 2008), which provides author profiles (affiliation, research terms, publication count, citation count, $h$-index) together with a co-authorship file recording collaboration counts between author pairs. To obtain a connected sub-network with a realistic, non-degenerate interference structure, we sample authors by snowball (breadth-first) expansion over the global co-authorship graph: starting from a seed author of moderate degree, we iteratively add collaborators until reaching $n = 2828$ authors. This procedure guarantees that the resulting co-authorship view is connected and that node degrees follow a heavy-tailed distribution typical of real collaboration networks, rather than the near-complete or disconnected graphs produced by random author selection.

We build $K = 3$ views for the sampled authors: **(A1) Co-authorship.** The observed relational network; an edge $(i, j)$ is present iff authors $i$ and $j$ have co-authored at least one publication. **(A2) Affiliation similarity** and **(A3)**

**Keyword (research-term) similarity.** Following Assumption 1, both are latent similarity graphs built solely from pre-treatment covariates. We embed each author's affiliation string and concatenated research terms with a pre-trained BERT encoder (`bert-base-uncased`, 768-dimensional `[CLS]` representation) (Devlin et al., 2019). For each view we compute pairwise cosine similarity and connect every node to its 10 nearest neighbours, then symmetrise and remove self-loops. The per-view statistics are reported in Appendix C.

Treatments and potential outcomes are simulated from Eq. (23) using the three views, with $\alpha_1 = \alpha_2 = 1$, $\alpha_3 = 0.5$ and $\varepsilon_i \sim \mathcal{N}(0,1)$, consistent with the protocol used for BlogCatalog and Flickr. The resulting co-authorship view satisfies the bounded-degree-growth condition required by our asymptotic theory.

## E   Theory

*Proof of Lemma 4.1.* We establish the static target; the stochastic target is obtained by replacing the point intervention $v_0$ with the kernel $h^*(\cdot \mid w)$ in (26).

Fix $v_0 \in \mathrm{supp}(V)$ and define the potential outcome $Y_i^*(v_0) := f_Y(v_0, W_i, \epsilon_{Y_i})$, so that $\psi_{\mathrm{stat}} = \frac{1}{n} \sum_{i=1}^n \mathbb{E}[Y_i^*(v_0)]$. By Eq. (2) there is a measurable map $g$ with $(V_i, W_i) = g(\epsilon_X, \epsilon_T)$; since $\epsilon_{Y_i} \perp (\epsilon_X, \epsilon_T)$ by Assumption 2, measurability yields

$$\epsilon_{Y_i} \perp (V_i, W_i). \tag{24}$$

For every $(v, w)$ with $\Pr(V_i = v, W_i = w) > 0$, guaranteed on $\mathrm{supp}(V) \times \mathrm{supp}(W)$ by Assumption 5, define $m(v,w) := \mathbb{E}[Y_i \mid V_i = v, W_i = w]$. Then

$$\begin{align}
m(v,w) &= \mathbb{E}[f_Y(v, w, \epsilon_{Y_i}) \mid V_i = v, W_i = w] && \text{(Eq. (2))} \\
&= \mathbb{E}[f_Y(v, w, \epsilon_{Y_i})] && \text{(by (24))} \\
&= \mathbb{E}[Y_i^*(v) \mid W_i = w], && \text{(Step 1; (24))} \tag{25}
\end{align}$$

and by Assumption 3 the law of $\epsilon_{Y_i}$ is common across units, so $m$ does not depend on $i$. By Assumption 1, $W_i$ is a function of pre-treatment quantities alone, hence its law $p_W$ is invariant under interventions on $V$. Combining with (25) and the tower property,

$$\psi_{\mathrm{stat}} = \frac{1}{n} \sum_{i=1}^n \mathbb{E}[\mathbb{E}[Y_i^*(v_0) \mid W_i]] \overset{(25)}{=} \frac{1}{n} \sum_{i=1}^n \mathbb{E}[m(v_0, W_i)] = \int m(v_0, w) \, p_W(dw). \tag{26}$$

The right-hand side depends on the observed law $P(W, V, Y)$ only. Steps (24)–(26) invoke solely the measurability of $g$, Assumptions 2–3, positivity (Assumption 5), and the pre-treatment property of $W$ (Assumption 1); none references the summaries $s_X^{(k)}, s_T^{(k)}$, the aggregators $\mathrm{Agg}_W, \mathrm{Agg}_V$, or the number of views $K$. Hence the identifying functional (26) is invariant across admissible aggregations, with only $\mathrm{supp}(V, W)$ changing. Assumption 4 is not used; it enters only in Theorem 4.1. □

*Proof of Proposition 4.1.* Under the true data-generating process $Y_i = f_Y(V_i^{\mathrm{true}}, W_i, \epsilon_{Y_i})$ with $V_i^{\mathrm{true}}$ pre-treatment, Assumption 2 gives $\epsilon_{Y_i} \perp (V_i, V_i^{\mathrm{true}}, W_i)$. Writing $\mu(v, w) = \mathbb{E}[f_Y(v, w, \epsilon_Y)]$, the same argument as in (25) yields, for $p_{(V,W)}$-a.e. $(v, w)$,

$$m(v,w) = \mathbb{E}\big[\mu(V^{\mathrm{true}}, w) \mid V = v, W = w\big], \tag{27}$$

$$\big|m(v,w) - \mu(v,w)\big| \leq \mathbb{E}\big[\big|\mu(V^{\mathrm{true}}, w) - \mu(v,w)\big| \,\big|\, V = v, W = w\big] \qquad \text{(Jensen)}$$

$$\leq L \, \mathbb{E}[|\delta| \mid V = v, W = w], \qquad \text{($L$-Lipschitz)} \tag{28}$$

where $\delta = V^{\mathrm{true}} - V$. *Static target:* setting $v = v_0$ in (28) and integrating against $p_W$,

$$\big|\psi_{\mathrm{stat}} - \psi_{\mathrm{stat}}^{\mathrm{true}}\big| \leq \int \big|m(v_0, w) - \mu(v_0, w)\big| \, p_W(dw) \leq L \int \mathbb{E}[|\delta| \mid V = v_0, W = w] \, p_W(dw). \tag{29}$$

*Stochastic target:* integrating (28) against $h^*(dv \mid w)\, p_W(dw)$ and changing measure via $h^*(dv \mid w) = \frac{h^*(v|w)}{h(v|w)}\, h(dv \mid w)$ (well-defined by Assumption 5, with $\frac{h^*}{h} \le \|\bar{h}^*/\bar{h}\|_\infty$),

$$\left|\psi_{\text{stoch}} - \psi_{\text{stoch}}^{\text{true}}\right| \le L \iint \mathbb{E}[|\delta| \mid V = v, W = w]\, \frac{h^*(v \mid w)}{h(v \mid w)}\, h(dv \mid w)\, p_W(dw)$$

$$\le L \left\|\tfrac{\bar{h}^*}{\bar{h}}\right\|_\infty \mathbb{E}[\mathbb{E}[|\delta| \mid V, W]] = L \left\|\tfrac{\bar{h}^*}{\bar{h}}\right\|_\infty \frac{1}{n} \sum_{i=1}^{n} \mathbb{E}|\delta_i|, \tag{30}$$

by the tower property. Finally, decomposing $\hat{\psi}_n - \psi^{\text{true}} = (\hat{\psi}_n - \psi) + (\psi - \psi^{\text{true}})$, the first term is $O_p(C_n^{-1/2})$ by Theorem 4.1 and the second is the deterministic bound above; consistency for $\psi^{\text{true}}$ holds when $\frac{1}{n}\sum_i \mathbb{E}|\delta_i| \to 0$, and the CAN statement transfers when it is $o(C_n^{-1/2})$. □

**Condition 1** (Bounded dependence / degree growth)**.** *The maximum degree of the union support graph $\tilde{A}$ grows slowly with sample size:*
$$K_{\max,n}^2/n \to 0$$
*(equivalently, $n/K_{\max,n}^2 \to \infty$, so that $C_n \to \infty$ in Theorem 4.1).*

This ensures that each unit's dependence neighborhood does not become too large relative to the overall sample, permitting a central limit theorem under Stein's method (Ogburn et al., 2022).

**Condition 2.** *Either the outcome regression $m(v, w)$ or the exposure mechanism $h(v \mid w)$ is consistently estimated at a sufficiently fast rate, or their estimation errors satisfy*
$$\|\hat{m} - m\| \cdot \|\hat{h} - h\| = o_p(n^{-1/2}).$$

*This guarantees that the second-order remainder in the influence function expansion vanishes.*

**Proposition E.1** (EIF form is invariant; only $m, \bar{h}$ change)**.** *Under Assumptions 1–3 and positivity, the efficient influence functions are*

$$D_{\text{stat}}^*(O) = \frac{\mathbb{1}\{V = v\}}{\bar{h}(v \mid W)}\{Y - m(v, W)\} + \big(m(v, W) - \psi_{\text{stat}}\big),$$

$$D_{\text{stoch}}^*(O) = \frac{\bar{h}^*(V \mid W)}{\bar{h}(V \mid W)}\{Y - m(V, W)\} + \Big(\int m(v, W) h^*(dv \mid W) - \psi_{\text{stoch}}\Big),$$

*with expectations $0$ and finite variances. The expressions are identical to the single-view case, with $m, \bar{h}$ computed for the aggregated $(W, V)$.*

**Lemma E.1** (Dependence and effective sample size under aggregation)**.** *Let $A^{(k)}$ have maximum degree $K_{\max,n}^{(k)}$, and recall the union support graph $\tilde{A}$ from Section 3.1. Then*

$$K_{\max,n} \le \sum_{k=1}^{K} K_{\max,n}^{(k)}.$$

*Under Assumption 4 (two-hop local dependence in $\tilde{A}$), the dependency graph for $\{D^*(O_i)\}_{i=1}^n$ has maximum degree*

$$\Delta_n \lesssim K_{\max,n}^2 \le \Big(\sum_{k=1}^{K} K_{\max,n}^{(k)}\Big)^2.$$

By Lemma 4.1 and Proposition E.1, the multi–view reparametrization preserves both the target functional and the EIF, so the EIF coincides with the forms in Ogburn et al. (2020). Hence, under the stated regularity (positivity, bounded moments) and $K_{\max,n}^2/n \to 0$, the CAN result can be obtained.

**Proposition E.2** (CLT with multi-view aggregation)**.** *Let $Z_i := D^*(O_i)$ with $\mathbb{E}|Z_i|^3 < \infty$. Assume $K_{\max,n}^2/n \to 0$. Then, with $C_n := n/(1 + \Delta_n)$ and $\Delta_n \lesssim K_{\max,n}^2$,*

$$\sqrt{C_n}\,(\mathbb{P}_n - P_0)D^*(O) \Rightarrow \mathcal{N}(0, \sigma^2), \qquad \sigma^2 = \text{Var}\{D^*(O)\}.$$

**Theorem E.1** (CAN with multi-view aggregation)**.** *Under Assumptions 1–5, Proposition E.1, the CLT of Proposition E.2, and the nuisance-rate condition $\|\hat{m} - m\|\,\|\hat{\bar{h}} - \bar{h}\| = o_p(C_n^{-1/2})$, the TMLE $\hat{\psi}_n$ for either target satisfies*

$$\sqrt{C_n}\,(\hat{\psi}_n - \psi) \Rightarrow \mathcal{N}(0, \sigma^2), \qquad C_n = \frac{n}{1 + \Delta_n}, \quad \Delta_n \lesssim K_{\max,n}^2.$$

Table 3: Ablation study on Flickr. MVDR-view$k$ uses only the $k$-th view; MVDR-noHSIC disables HSIC regularisation. All variants use identical hyperparameters. Best result in each column is **bolded**.

| Method | ATE-indiv | ATE-total | PEHE-indiv | PEHE-total |
|---|---|---|---|---|
| MVDR-N1 | $0.2038_{\pm 0.1099}$ | $0.2153_{\pm 0.1297}$ | $3.0374_{\pm 0.0579}$ | $3.0175_{\pm 0.0856}$ |
| MVDR-N2 | $0.2212_{\pm 0.1508}$ | $0.2382_{\pm 0.1533}$ | $3.0459_{\pm 0.0591}$ | $3.0284_{\pm 0.0716}$ |
| MVDR-N3 | $0.2091_{\pm 0.1132}$ | $0.2156_{\pm 0.1046}$ | $3.0366_{\pm 0.0619}$ | $3.0420_{\pm 0.0875}$ |
| MVDR-noHSIC | $0.1632_{\pm 0.1053}$ | $0.1783_{\pm 0.2052}$ | $3.0373_{\pm 0.0532}$ | $3.0273_{\pm 0.0698}$ |
| MVDR (full) | $\mathbf{0.1623}_{\pm 0.1050}$ | $\mathbf{0.1782}_{\pm 0.1800}$ | $\mathbf{3.0354}_{\pm 0.0547}$ | $\mathbf{3.0261}_{\pm 0.0711}$ |

# F  Additional Experimental Results

**Implementation details.** All GCN layers use hidden dimension 64. The attention fusion module uses a single linear layer. The soft histogram density estimator uses $C = 10$ bins. The B-spline perturbation network uses degree-3 splines with 10 knots. All models are trained with Adam optimizer (learning rate $10^{-3}$) for 500 epochs. Results are reported as mean $\pm$ standard deviation over 5 independent runs with different random seeds.

## F.1  Ablation Study

**Ablation: contribution of multi-view aggregation.** To directly assess the benefit of aggregating multiple network views, we compare MVDR against three single-view variants (MVDR-n1, MVDR-n2, MVDR-n3) and one variant with the HSIC regularisation disabled (MVDR-noHSIC), all trained with identical hyperparameters. Results on Flickr are reported in Table 3. On the Flickr dataset, MVDR (full) outperforms all single-view variants in 4 evaluation metrics, demonstrating that aggregating complementary information from multiple network views consistently improves causal effect estimation. This suggests that while each individual view captures some useful structural information, the full multi-view representation provides complementary signals that no single view alone can recover. Removing the HSIC regularisation (MVDR-noHSIC) also leads to degradation in 4 metrics, confirming that encouraging diversity across view-specific representations is beneficial.

## F.2  Additional Metrics

The main purpose of this paper is to evaluate the average potential outcome $E[\bar{Y}_n^*]$, which means the outcome for an observed network with latent dependency if certain hypothetical intervention is assigned to it. In Table 4, we also report the results for average main effect, average spillover effect, average total effect and individual main effect, individual spillover effect, individual total effect by pre-defining the exposure mapping $v$ as the average ratio of treated neighbors across views.

Define $\psi_i$ as $\psi_i(t, z) = m_t(z, w_i) + \epsilon(z, w_i) \cdot H_i$, and $\psi$ as $\psi(t, z) = \frac{1}{n} \sum_{i=1}^{n} \psi_i(t, z)$. Then we have the following definitions:

- The average main effect (AME) measures the difference in the mean outcomes between units assigned to $T = t, Z = 0$ and assigned to $T = t', z = 0$: $\tau^{(t,0),(t',0)} = \psi(t, 0) - \psi(t', 0)$.

- The average spillover effect (ASE) measures the difference in mean outcomes between units assigned to $T = 0, Z = z$ and assigned to $T = 0, Z = z'$: $\tau^{(0,z),(0,z')} = \psi(0, z) - \psi(0, z')$.

- The average total effect (ATE) measures the difference in mean outcomes between units assigned to $T = t, Z = z$ and assigned $T = t', Z = z'$: $\tau^{(t,z),(t',z')} = \psi(t, z) - \psi(t', z')$.

- The individual main effect (IME) measures the difference in mean outcomes of a particular unit $x_i$ assigned to $T = t, Z = 0$ and assigned $T = t', Z = 0$: $\tau_i^{(t,0),(t',0)} = \psi_i(t, 0) - \psi_i(t', 0)$.

- The individual spillover effect (ISE) measures the difference in mean outcomes of a particular unit $x_i$ assigned to $T = 0, Z = z$ and assigned $T = 0, Z = z'$: $\tau_i^{(t,z),(t',z')} = \psi_i(0, z) - \psi_i(0, z')$.

- The individual total effect (ITE) measures the difference in mean outcomes of a particular unit $x_i$ assigned to $T = t, Z = z$ and assigned $T = t', Z = z'$: $\tau_i^{(t,z),(t',z')} = \psi_i(t, z) - \psi_i(t', z')$.

Table 4: Comparison of estimation errors across different models and datasets. Best performance in each row is **bolded**. '−' represents the results are not reported by that method.

| Split | Metric | dataset | effect | CFR | CFR+z | ND | ND+z | TARNet | TARNet+z | NetEst | HINITE | TNet | MVDR |
|---|---|---|---|---|---|---|---|---|---|---|---|---|---|
| With-in-Sample | $\varepsilon_{average}$ | BlogCata | AME | $\mathbf{0.1425}_{0.1516}$ | $0.2549_{0.1329}$ | $0.1590_{0.0859}$ | $0.3080_{0.1620}$ | $0.1550_{0.0683}$ | $0.3335_{0.1443}$ | $0.3533_{0.1410}$ | − | $40.3076_{92.4736}$ | $0.1720_{0.0995}$ |
| | | | ASE | $0.1735_{0.1516}$ | $0.2509_{0.2208}$ | $0.1735_{0.1516}$ | $0.2430_{0.2278}$ | $\mathbf{0.1734}_{0.0696}$ | $0.2241_{0.2090}$ | $0.2191_{0.2231}$ | − | $145.1494_{214.8604}$ | $0.1905_{0.1016}$ |
| | | | ATE | $0.2449_{0.1704}$ | $0.1650_{0.0786}$ | $0.1326_{0.1208}$ | $\mathbf{0.1105}_{0.1060}$ | $0.1326_{0.1076}$ | $0.1308_{0.1385}$ | $0.5076_{0.1546}$ | $0.6230_{0.4354}$ | $5707.6897_{9251.3347}$ | $\mathbf{0.1639}_{0.1457}$ |
| | | Flickr | AME | $0.2691_{0.1097}$ | $0.1692_{0.1467}$ | $0.2726_{0.2432}$ | $0.4251_{0.1537}$ | $0.2918_{0.2166}$ | $0.3492_{0.0897}$ | $0.7693_{0.0524}$ | − | $12.0787_{11.3391}$ | $\mathbf{0.0593}_{0.0470}$ |
| | | | ASE | $\mathbf{0.0887}_{0.0211}$ | $0.1767_{0.0620}$ | $0.1078_{0.0822}$ | $0.1964_{0.0883}$ | $0.1078_{0.0822}$ | $0.2736_{0.0700}$ | $0.1948_{0.0679}$ | − | $158.5197_{175.1475}$ | $0.2296_{0.1129}$ |
| | | | ATE | $0.2617_{0.1099}$ | $0.1887_{0.1861}$ | $0.3048_{0.1872}$ | $0.3013_{0.1364}$ | $0.2833_{0.2067}$ | $0.3701_{0.1785}$ | $1.0245_{0.1374}$ | $\mathbf{0.1751}_{0.1004}$ | $2923.5093_{3001.1783}$ | $0.4908_{0.2424}$ |
| | | DBLP | AME | $0.2153_{0.1557}$ | $0.2606_{0.1441}$ | $0.5965_{0.2617}$ | $0.2916_{0.1325}$ | $1.0216_{0.6321}$ | $0.3672_{0.2286}$ | $0.8758_{0.2556}$ | − | $3.5683_{11.7916}$ | $\mathbf{0.0336}_{0.0348}$ |
| | | | ASE | $0.2606_{0.1769}$ | $0.329_{0.2398}$ | $0.3108_{0.1768}$ | $0.3311_{0.1755}$ | $0.1381_{0.0438}$ | $\mathbf{0.1376}_{0.0429}$ | $0.2619_{0.1827}$ | − | $7.5224_{11.0523}$ | $\mathbf{0.2611}_{0.0314}$ |
| | | | ATE | $\mathbf{0.076}_{0.0833}$ | $0.1739_{0.0943}$ | $0.8423_{0.2742}$ | $0.4265_{0.0574}$ | $1.1592_{0.5784}$ | $0.2585_{0.1698}$ | $1.0077_{0.0836}$ | $0.6815_{0.3553}$ | $11.982_{16.2407}$ | $0.4408_{0.0527}$ |
| | $\varepsilon_{individual}$ | BlogCata | IME | $3.0597_{0.1232}$ | $3.0358_{0.1232}$ | $2.9224_{0.094}$ | $2.9359_{0.0842}$ | $2.9427_{0.0894}$ | $2.9570_{0.0808}$ | $2.9412_{0.0865}$ | − | $140.5076_{127.1403}$ | $\mathbf{2.8966}_{0.0583}$ |
| | | | ISE | $2.8973_{0.0845}$ | $2.9072_{0.0845}$ | $2.8973_{0.0852}$ | $2.9069_{0.0842}$ | $2.8973_{0.0826}$ | $2.9044_{0.0836}$ | $2.9088_{0.0800}$ | − | $270.6221_{341.4499}$ | $\mathbf{2.7668}_{0.0404}$ |
| | | | ITE | $3.0540_{0.097}$ | $3.0182_{0.0970}$ | $2.9004_{0.0510}$ | $2.8967_{0.0496}$ | $2.9216_{0.0529}$ | $2.9185_{0.0591}$ | $2.9353_{0.0535}$ | $3.3113_{0.2458}$ | $12038.2843_{15704.8492}$ | $\mathbf{2.8692}_{0.0482}$ |
| | | Flickr | IME | $9.3637_{5.1163}$ | $2.9457_{0.0596}$ | $6.0954_{1.8223}$ | $2.9607_{0.0690}$ | $2.9683_{0.0648}$ | $2.9124_{0.0499}$ | $2.9537_{0.0498}$ | − | $35.5421_{32.8715}$ | $\mathbf{0.6581}_{0.2529}$ |
| | | | ISE | $2.8980_{0.0439}$ | $2.8965_{0.0976}$ | $3.0014_{0.0237}$ | $3.0342_{0.0964}$ | $2.9228_{0.0547}$ | $2.8865_{0.0989}$ | $2.8896_{0.1010}$ | − | $440.6491_{463.7988}$ | $\mathbf{0.6886}_{0.2660}$ |
| | | | ITE | $9.3395_{5.1269}$ | $3.0072_{0.641}$ | $6.1200_{1.8291}$ | $2.9525_{0.0830}$ | $2.9506_{0.0622}$ | $2.9470_{0.0449}$ | $3.0514_{0.0463}$ | $3.0427_{0.0662}$ | $9314.1054_{9633.2401}$ | $0.8652_{0.2978}$ |
| | | DBLP | IME | $3.3294_{0.2079}$ | $3.3316_{0.2094}$ | $3.0448_{0.0829}$ | $2.6218_{0.0341}$ | $2.8543_{0.2530}$ | $2.6366_{0.0541}$ | $3.4669_{0.2156}$ | − | $6.3263_{11.4395}$ | $\mathbf{0.0506}_{0.0353}$ |
| | | | ISE | $3.4736_{0.1455}$ | $3.483_{0.1517}$ | $2.9793_{0.0928}$ | $2.6923_{0.0325}$ | $2.6921_{0.0321}$ | $2.6921_{0.0322}$ | $3.508_{0.1563}$ | − | $12.3642_{21.6718}$ | $\mathbf{0.2852}_{0.0557}$ |
| | | | ITE | $3.3944_{0.0849}$ | $3.3985_{0.0839}$ | $3.0996_{0.0805}$ | $2.5758_{0.0772}$ | $3.2055_{0.1500}$ | $2.5984_{0.1022}$ | $3.5871_{0.0966}$ | $2.9779_{0.2013}$ | $13.5564_{15.9518}$ | $\mathbf{0.4881}_{0.0460}$ |
| Out-of-Sample | $\varepsilon_{ATE}$ | BlogCata | AME | $0.2103_{0.154}$ | $0.203_{0.1861}$ | $0.5117_{0.2551}$ | $0.6367_{0.3633}$ | $0.4508_{0.3217}$ | $0.5380_{0.4065}$ | $0.2312_{0.2088}$ | − | $40.3076_{92.4736}$ | $\mathbf{0.1289}_{0.0416}$ |
| | | | ASE | $0.1277_{0.0965}$ | $0.1882_{0.1349}$ | $0.1277_{0.0965}$ | $0.1825_{0.1322}$ | $0.1277_{0.0965}$ | $0.1736_{0.1373}$ | $\mathbf{0.1209}_{0.098}$ | − | $145.1494_{214.8604}$ | $0.1678_{0.0751}$ |
| | | | ATE | $0.2449_{0.1704}$ | $0.2726_{0.2062}$ | $0.5484_{0.3592}$ | $0.7826_{0.4412}$ | $0.4874_{0.2824}$ | $0.6466_{0.3515}$ | $0.3724_{0.2632}$ | $0.3574_{0.2483}$ | $5707.6897_{9251.3347}$ | $\mathbf{0.1639}_{0.1055}$ |
| | | Flickr | AME | $0.2691_{0.1097}$ | $0.2908_{0.2120}$ | $0.1662_{0.1123}$ | $0.4614_{0.2634}$ | $\mathbf{0.1626}_{0.1219}$ | $0.4107_{0.2165}$ | $0.7582_{0.1641}$ | − | $9.0943_{10.9997}$ | $0.2605_{0.0590}$ |
| | | | ASE | $0.0887_{0.0211}$ | $0.2074_{0.0858}$ | $\mathbf{0.0887}_{0.0211}$ | $0.2475_{0.1138}$ | $0.0887_{0.0211}$ | $0.2106_{0.1111}$ | $0.1406_{0.1115}$ | − | $92.6766_{87.5788}$ | $0.2426_{0.0766}$ |
| | | | ATE | $0.2617_{0.1099}$ | $\mathbf{0.1940}_{0.1716}$ | $0.2669_{0.1859}$ | $0.2878_{0.1566}$ | $0.3395_{0.1229}$ | $0.2618_{0.1089}$ | $0.7807_{0.1527}$ | $0.1674_{0.1618}$ | $2152.0205_{1924.3221}$ | $0.5516_{0.3424}$ |
| | | DBLP | AME | $0.2153_{0.1557}$ | $0.2606_{0.1441}$ | $0.7047_{0.2382}$ | $0.284_{0.1825}$ | $0.8104_{0.4552}$ | $0.4669_{0.1738}$ | $0.8758_{0.2556}$ | − | $3.5683_{11.7916}$ | $\mathbf{0.0506}_{0.0353}$ |
| | | | ASE | $\mathbf{0.2606}_{0.1769}$ | $0.329_{0.2398}$ | $0.2606_{0.1769}$ | $0.2718_{0.1793}$ | $0.2606_{0.1769}$ | $0.2627_{0.176}$ | $0.2619_{0.1827}$ | − | $7.5224_{11.0523}$ | $0.2852_{0.0557}$ |
| | | | ATE | $\mathbf{0.076}_{0.0833}$ | $0.1739_{0.0943}$ | $0.8558_{0.0891}$ | $0.3097_{0.1377}$ | $0.9616_{0.325}$ | $0.5847_{0.1165}$ | $1.0077_{0.0836}$ | $0.7259_{0.4971}$ | $11.982_{16.2407}$ | $\mathbf{0.4881}_{0.0460}$ |
| | $\varepsilon_{PEHE}$ | BlogCata | IME | $3.6311_{1.2841}$ | $3.9781_{1.7998}$ | $5.666_{2.2252}$ | $7.2773_{3.4133}$ | $4.7532_{2.3242}$ | $5.5978_{4.2786}$ | $3.1134_{0.1372}$ | − | $140.5076_{127.1403}$ | $\mathbf{2.9013}_{0.0460}$ |
| | | | ISE | $2.9072_{0.0845}$ | $2.8033_{0.0519}$ | $2.7983_{0.0453}$ | $2.8027_{0.0516}$ | $2.7983_{0.0453}$ | $2.8022_{0.0522}$ | $3.0264_{0.124}$ | − | $270.6221_{341.4499}$ | $\mathbf{2.7966}_{0.0467}$ |
| | | | ITE | $3.6261_{1.255}$ | $3.9765_{1.7725}$ | $5.6698_{2.2135}$ | $7.2954_{3.3815}$ | $4.7462_{2.3115}$ | $5.5797_{4.2738}$ | $3.0796_{0.1691}$ | $3.2063_{0.3024}$ | $12038.2843_{15704.8492}$ | $\mathbf{2.8580}_{0.0460}$ |
| | | Flickr | IME | $9.3637_{5.1163}$ | $3.573_{0.6824}$ | $4.0064_{1.4430}$ | $3.1551_{0.1926}$ | $3.2343_{0.1757}$ | $3.0985_{0.0172}$ | $3.0916_{0.0363}$ | − | $68.2053_{39.6463}$ | $\mathbf{0.7660}_{0.2204}$ |
| | | | ISE | $3.0457_{0.0475}$ | $3.0295_{0.0976}$ | $3.0457_{0.0475}$ | $3.0566_{0.0409}$ | $3.0457_{0.0475}$ | $3.0314_{0.0964}$ | $3.0338_{0.0290}$ | − | $352.7498_{303.2289}$ | $\mathbf{0.7214}_{0.2673}$ |
| | | | ITE | $9.3395_{5.1269}$ | $3.5895_{0.641}$ | $4.0251_{1.4578}$ | $3.1328_{0.1616}$ | $3.242_{0.1979}$ | $3.0533_{0.0709}$ | $3.0834_{0.0812}$ | $3.1153_{0.0732}$ | $8165.584_{7729.401}$ | $\mathbf{0.9543}_{0.3456}$ |
| | | DBLP | IME | $3.3294_{0.2079}$ | $3.3316_{0.2094}$ | $3.4027_{0.1876}$ | $3.3357_{0.2167}$ | $3.45_{0.1486}$ | $3.3558_{0.2131}$ | $3.4669_{0.2156}$ | − | $6.3263_{11.4395}$ | $\mathbf{0.5119}_{0.0155}$ |
| | | | ISE | $3.4736_{0.1455}$ | $3.483_{0.1517}$ | $3.4736_{0.1455}$ | $3.4745_{0.146}$ | $3.4736_{0.1455}$ | $3.4737_{0.1455}$ | $3.508_{0.1563}$ | − | $12.3642_{21.6718}$ | $\mathbf{0.5175}_{0.0370}$ |
| | | | ITE | $3.3944_{0.0849}$ | $3.3985_{0.0839}$ | $3.5013_{0.0936}$ | $3.4116_{0.0815}$ | $3.5403_{0.1502}$ | $3.4462_{0.0883}$ | $3.5871_{0.0966}$ | $0.8227_{0.219}$ | $13.5564_{15.9518}$ | $\mathbf{0.7080}_{0.0249}$ |

For the average effects, we use mean absolute error (MAE) as metric: $\epsilon_{average} = |\hat{\tau} - \tau|$. For the individual effects, we use Precision in Estimation of Heterogeneous Effect (PEHE) as metric: $\epsilon_{individual} = \frac{1}{n}\sum_{i=1}^{n}(\tau_i - \hat{\tau}_i)^2$. Specifically, HINITE does not identify spillover effects and main effects, so the results are not reported.

