# OpenReview forum: "A Targeted Learning Framework for Policy Evaluation with Unobserved Network Interference"
_TMLR — Under review for TMLR_

### Review · Reviewer_Jfns · 2026-02-03

**Summary Of Contributions:**

The paper suggests a new method for estimating average potential outcomes under interventions (policy value) in the network interference setting. More specifically, the authors consider a setting where multiple views of the ground-truth adjacency matrix are available. The final method, called a multi-view doubly robust estimator (Mvdr), relies on the targeted maximum likelihood estimation (TMLE) framework and, thus, possesses the property of double-robustness and semi-parametric efficiency. Finally, the authors compared their method with other baselines adapted to their setting based on several semi-synthetic datasets.

Key strength:
1. The setting, considered in the paper, is quite unique and under-explored
2. The proposed estimator has favourable theoretical properties

Key weaknesses:
1. **Confusing setting**. I struggled to fully understand two main aspects of the proposed setting: (1) summary variables and (2) multi-view data. (1) It wasn't fully clear whether the summary variables are learned or given in the data. (2) I don't fully understand how the identification and estimation results relate to the fact that we observe multiple views of the ground-truth matrix A.

**Additional Comments:**

- I don't understand why there are two causal diagrams in Fig.1 (and how they differ)
- Is there a justification on why the coefficient of the clever covariate $\varepsilon$ depends on $v$ and $w$? To the best of my understanding, the TMLE estimators for finite-dimensional quantities (e.g., policy values [1]) require fitting the constant $\varepsilon$.


References:
- [1] Muñoz, Iván Díaz, and Mark Van Der Laan. "Population intervention causal effects based on stochastic interventions." Biometrics 68.2 (2012): 541-549.

**Audience:**

Yes

**Audience Explanation:**

I think this paper (when all the weaknesses above are addressed) can serve as an important baseline in the network interference setting.

**Claims And Evidence:**

No

**Claims Explanation:**

- Authors claimed they propose a new identitication results. Yet, I found standard assumptions under network interference and standard identification formulas (= Sec. 3). Is the main novelty in Lemma C.1?
-  Given weakness 1, the design of the synthetic experiments is not fully clear. Namely, it is not clear whether the Mvdr takes several graphs A^(m) as the input or just one.

**Requested Changes:**

Major:
- I encourage the authors clarify the weakness 1.
- I think the paper would profit from having a formula for the TMLE estimator already in Sec. 4.1


Minor:
- The first paragraph of the Introduction reads rather as Related Work.
- Some of the notation is inconsistent. For example, $\varepsilon_{C_i}$ is not defines, $M$ should be caligraphic before Eq. (5).

---

> ### Author Response · Authors · 2026-03-16
> **Response to Reviewer jfns**
>
> We thank the reviewer for their thorough and constructive evaluations. We have carefully addressed all comments and revised the manuscript accordingly. Below we provide a point-by-point response to each raised concern. All changes in the revised manuscript are highlighted in blue.
>
> ---
>
> **[Major 1] Clarification of summary variables and multi-view data**
>
> **(1) Learned vs. pre-specified summaries.** The summary functions $s_X, s_T$ are **learned** from data via GCNs, not pre-specified. This is clarified in new Remark 1 (Section 3.4). The theoretical justification is Lemma 1 (Section 4.1): the EIF retains its canonical form for *any* Borel-measurable aggregation satisfying Assumptions 1–3, so data-adaptive learning of $(W,V)$ does not invalidate identification or double robustness.
>
> **(2) Role of multi-view networks.** Our setting differs from Lin et al. (2023) (HINITE), where all views are directly observed. In our framework, $A^{(1)}$ is the **directly observed** network; $A^{(2)}, A^{(3)}$ are **proxy networks constructed from pre-treatment covariates** (KNN similarity, per Assumption 1). This models the realistic scenario where the true interference network is only partially observed.
>
> In all experiments, **Mvdr takes all three matrices $\{A^{(1)}, A^{(2)}, A^{(3)}\}$ simultaneously**. Each view is processed by a separate GCN; embeddings are fused via attention into $(W_i, V_i)$. The multi-view construction affects estimation (through $s_X, s_T$) but not identification — Lemma 1 guarantees the EIF form is invariant to the aggregation rule.
>
> The new ablation (Table 3, Flickr) directly confirms the benefit: Mvdr (full) outperforms all single-view variants in 4 metrics.
>
> ---
>
> **[Major 2] Novelty of identification results**
>
> The reviewer is correct: Section 3 uses standard identification arguments. The novelty is **Lemma 1** (promoted from Appendix C to Section 4.1):
>
> > **(Lemma 1, EIF invariance under multi-view aggregation.)** Let $(W,V)$ be any Borel-measurable aggregation of $(X,T,A^{(1)},\ldots,A^{(M)})$ with $W$ depending only on pre-treatment inputs. Under Assumptions 1–3, $\psi_{\mathrm{stat}}$ and $\psi_{\mathrm{stoch}}$ are identified from $P(W,V,Y)$ and the EIF takes the canonical forms of Eqs. (5)–(6).
>
> *Proof sketch.* Each $A^{(m)}$ is pre-treatment (Assumption 1), so $(W_i,V_i)$ are valid adjustment variables. The target functional depends on $(W,V)$ only through $m(v,w)$ and $h(v|w)$; the pathwise derivative is invariant to the aggregation rule. $\square$
>
> **Implication:** TMLE is doubly robust for *any* admissible $(W,V)$ including GNN-learned ones — prior work (Chen et al., 2024) requires the exposure mapping to be pre-specified.
>
> ---
>
> **[Major 3] TMLE formula in Section 4.1**
>
> Added as a 4-step algorithm:
> 1. Estimate $\hat{m}(v,w)$ and $\hat{h}(v|w)$.
> 2. Compute $H_i = \bar{h}^*(v_i,w_i)/\bar{h}(v_i,w_i)$.
> 3. Fit $\epsilon(v)$ via influence function loss: $\tilde{Y}^*_i = \hat{m}(v^*_i,w_i) + \epsilon(v^*_i)\cdot H_i$.
> 4. $\hat{\psi} = \frac{1}{n}\sum_i \tilde{Y}^*_i$.
>
> ---
>
> **[Minor 1] Introduction reads as Related Work**
>
> Restructured: first paragraph now motivates the problem directly; literature review condensed and moved to Section 2.
>
> ---
>
> **[Minor 2] Inconsistent notation**
>
> $\bar{h}(v,w)=\frac{1}{n}\sum_j h_j(v,w)$ now defined before Eq. (4). $\mathcal{M}$ now calligraphic consistently. All undefined symbols introduced at first use.
>
> ---
>
> **[Additional Comment 1] Figure 1**
>
> Caption revised: left panel — our model with learned graph $G$ (proxy networks constructed from covariates per Assumption 1); right panel — standard SEM with fixed observed network.
>
> ---
>
> **[Additional Comment 2] Why $\epsilon$ depends on $v$**
>
> Standard TMLE fits scalar $\epsilon$ for discrete $V$. In our setting $V=(T,S)$ has a **continuous component** $S$ (neighbourhood treatment average), so the EIF ratio $\bar{h}^*(V|W)/\bar{h}(V|W)$ varies continuously with $v$. A scalar $\epsilon$ cannot correct position-varying bias. Following VCNet (Nie et al., 2021), we parameterise $\epsilon(v)$ as a B-spline network. For discrete $V$ (static intervention, Target A), $\epsilon$ reduces to a finite-dimensional vector, recovering the standard formulation.
>
> ---
> Ref:
>
> [1]Lin, Xiaofeng & Zhang, Guoxi & Lu, Xiaotian & Bao, Han & Takeuchi, Koh & Kashima, Hisashi. (2023). Estimating Treatment Effects Under Heterogeneous Interference. 10.1007/978-3-031-43412-9_34.
>
> [2]Weilin Chen, Ruichu Cai, Zeqin Yang, Jie Qiao, Yuguang Yan, Zijian Li, and Zhifeng Hao. Doubly robust causal effect estimation under networked interference via targeted learning, 2024. URL https://arxiv.org/abs/2405.03342.
>
> [3]Nie L, Ye M, Liu Q, et al. Vcnet and functional targeted regularization for learning causal effects of continuous treatments[J]. arXiv preprint arXiv:2103.07861, 2021.

---

> > ### Comment · Reviewer_Jfns · 2026-07-02
> >
> > I thank the authors for the revised version of the paper and for resolving some of the issues. However, some things remain unclear to me:
> > 1. I haven't found a new Remark 1 in the revised manuscript.
> > 2. The proof of the new identification statement (Lemma 1) is not fully clear to me. I encourage the authors to provide a more formal, step-by-step proof (e.g., by using Assumptions 1-5).  For the same reason, I don't understand why "The multi-view construction affects estimation ... but not identification". Do the authors mean that the partial observability of the adjacency matrix is as good as random?
> > 3. Now I also got confused: why are there exactly 3 views, A^(1), A^(2), A^(3)  (two of which are learned by KNN)? Is it a general feature of the method (e.g., as suggested by Fig. 2)? Is a KNN part of Assumption 1?
> >
> > Yet, for now, the current version of the paper is more confusing for a reader than helpful: As noted by one of the other reviewers, "The manuscript does not provide sufficient details on how the multiple views are constructed".
> > Given that the main contribution seems to be a partially observable setting ("Our setting differs from Lin et al. (2023) (HINITE), where all views are directly observed."), I encourage the authors to carefully focus on the missingness mechanism and how that affects identifiability.

---

> > > ### Author Response · Authors · 2026-07-05
> > >
> > > We thank the reviewer for the follow-up and apologize that the promised remark was lost during manuscript revision. All changes are marked in blue.
> > >
> > > ---
> > >
> > > **Q1: Missing Remark 1.**
> > >
> > > Now in place as **Remark 1**, after Lemma 4.1: the summaries are learned via the GCN modules of Section 5; Lemma 4.1 licenses this because its proof uses only measurability of the aggregation and the pre-treatment property of $W$, so the identifying functionals (Lemma 4.1) and the EIFs (Proposition D.1) hold for any admissible aggregation and any $K$; $K=3$ is an experimental choice.
> > >
> > > ---
> > >
> > > **Q2: Formal proof.**
> > >
> > > **(a)** The proof (Appendix D) is now more formal, with each step tagged by the assumption it uses: display (24) gets $\epsilon_Y \perp (V, W)$ from Assumption 2; display (25) derives $m(v,w) = \mathbb{E}[Y^*(v) \mid W = w]$, using Assumption 5 (positive conditioning event) and Assumption 3 ($m$ common across units); display (26) aggregates via the tower property, using Assumption 1 for the intervention-invariance of $p_W$.
> > >
> > > **(b)** $\psi$ is defined on the constructed exposures $(V,W)$ — an expected exposure effect (Sävje, 2024), stated in Section 3.4. Lemma 4.1 identifies it for any pre-treatment aggregation, and Theorem 4.1 holds for it unconditionally; that is the precise version of the retired phrase. $\psi^{\rm true}$ is the same functional on the true exposure $V^{\rm true}$. We add the new Proposition 4.1 to bound it: with $\delta_i = V_i^{\rm true} - V_i$ and $\mu$ $L$-Lipschitz, $|\psi - \psi^{\rm true}| \le L \Vert \bar h^\ast / \bar h \Vert_\infty \cdot \frac{1}{n}\sum_{i=1}^n \mathbb{E}|\delta_i|$, so the bias is linear in the exposure error of the views. Assumption 6 (exact sufficiency) is the $\delta \equiv 0$ limit; the TMLE is consistent for $\psi^{\rm true}$ when $\frac{1}{n}\sum_i \mathbb{E}|\delta_i| \to 0$, and Theorem 4.1 transfers when this is $o(C_n^{-1/2})$.
> > >
> > > ---
> > >
> > > **Q3: Why 3 views? Is KNN part of Assumption 1?**
> > >
> > > $K$ is a hyperparameter and KNN is not part of Assumption 1, merely one admissible construction. $K=3$ follows the protocol of Lin et al. (2023) (one observed network plus two similarity views); details in Section 6 and Appendix C.
> > >
> > > ---
> > >
> > > **On the missingness mechanism.**
> > > $\psi$ is identified regardless of the missingness mechanism (Lemma 4.1); its distance to the true-network effect is controlled by $\mathbb{E}|\delta|$ (Proposition 4.1); with validation data on true edges, $\mathbb{E}|\delta|$ becomes partially estimable (Chao et al., 2025). The difference between HINITE and our method is not observability per se, but that our estimand remains well-defined and doubly robust when views are proxies.
> > >
> > > ---
> > >
> > > **References**
> > >
> > > [1] Sävje, F. Causal inference with misspecified exposure mappings: separating definitions and assumptions. *Biometrika*, 111(1):1–15, 2024.
> > >
> > > [2] Lin, X., Zhang, G., Lu, X., Bao, H., Takeuchi, K., Kashima, H. Estimating treatment effects under heterogeneous interference. *ECML PKDD*, 2023.
> > >
> > > [3] Chao, A., Spiegelman, D., Buchanan, A., Forastiere, L. Estimation and inference for causal spillover effects in egocentric-network randomized trials in the presence of network membership misclassification. *Biostatistics*, 26(1):kxaf009, 2025.

---

### Review · Reviewer_C1Je · 2026-02-10

**Summary Of Contributions:**

This manuscript focuses on the challenges that arise when we have multiple partially observed networks and are attempting to assess the causal impact of different policies.  This approach introduces new methods to summarize the input graph through a graph neural network and estimate densities to use doubly robust approaches.  Model comparisons suggest that the proposed approach works better in this particular setting.

**Audience:**

Yes

**Audience Explanation:**

Graph-based causal inference is a widely applied area of research.

**Broader Impact Concerns:**

N/A.

**Claims And Evidence:**

No

**Claims Explanation:**

There are several issues in this manuscript that need to be addressed.

First, the results do not include enough details to understand the full procedure, the model selection process, and the sources of variability.  Given the extremely high variability in the results, there is not a clear improvement from the proposed Mvdr method compared to baselines (the authors should consider some statistical tests).  Additionally, this high variability is unsatisfactorily explained.  Additionally, the comparisons are not explained in enough detail to explain why there is a performance gap.  The experimental setup needs to be explained in enough detail so that the mismatch from the assumptions of the baseline models and the data are clear.

Second, several of the claims appear imprecise, and need to be revised to address these challenges.  For example: "1. We develop a doubly robust estimator for causal inference on partially observed networks exhibiting latent network dependence. We prove that the estimator’s efficient influence function is invariant to the admissible summary mappings, and this EIF invariance yields double robustness."
It is not clear exactly what the authors are claiming as the contribution here.  Doubly robust estimators exist for graph structured data, and the procedure seems pretty much the same once the summary values are determined.  In my reading, it seems like the biggest change is on how we are summarizing data, not on on the theory or development of the doubly robust estimator.  If that is untrue, the claims need to be revised, and the sections on the doubly robust estimator need to more completely cover prior material and explain the changes.

Overall, the difference between this approach and prior approaches needs to be clearer, and that needs to be more completely explained (both discussion and ablations, etc) to show how these improvements arise.

**Requested Changes:**

Major:
1. Add necessary detail and explanations to the results, including full procedures, model selection, statistics, details on baseline models, and ablations.
2. Clarify contributions of the manuscript by being more explicit about what has changed from prior work.

Minor:
1. Assumption 4 seems more limited to the way it is described in the manuscript (e.g., partially observed networks).  In particular, this just says we can have neighbors of neighbors influence, but no further.  In some sense, just connecting neighbors of neighbors in the graph makes this so that only neighbors can influence.  It's not clear how much this assumption differs from prior assumptions rather than just a representational choice.
2.  Please clean up minor notational issues.  For example, be explicit about self-connections. In (1), note that the {} go over j. C is not introduced or explained in Assumption 1. Assumption 3 is unclear whether you mean that the errors are all identically distributed or each set separately. v_i* is never defined--while it is fairly straightforward from context, it should be explicitly defined.
3. Explain what we are supposed to take from Figure 1 on the left versus the right.
4. "Dn(o) will have an expected value of 0 at the true ψn." Since ψn has not been defined at that point, the definition needs to be redrafted.  Really ψn is defined to make D_N(o)=0.
5. Figure 2 is not fully defined with all of the abbreviations.  In particular, it's unclear what the network learner is doing, as elsewhere it seems the graphs are given.

---

> ### Author Response · Authors · 2026-03-16
> **Response to Reviewer C1Je**
>
> We thank the reviewer for their thorough and constructive evaluations. We have carefully addressed all comments and revised the manuscript accordingly. Below we provide a point-by-point response to each raised concern. All changes in the revised manuscript are highlighted in blue.
>
> **[Major 1] Experimental details, variance, and statistical tests**
>
> **(a) Setup clarification.** We have added Section 6.1 detailing: (i) Mvdr takes all $K=3$ adjacency matrices $\{A^{(1)}, A^{(2)}, A^{(3)}\}$ simultaneously — $A^{(1)}$ is the directly observed network; $A^{(2)}, A^{(3)}$ are proxy networks constructed from covariates via KNN similarity. Single-network baselines (CFR, ND, TARNet, TNet) receive only $A^{(1)}$. (ii) "+z" variants augment baselines with exposure summary $z_i=|N_i|^{-1}\sum_{j\in\mathcal{N}_i}T_j$.
>
> **(b) Baseline mismatch.** The performance gap has two sources. First, baselines only observe $A^{(1)}$ and miss the structural information in $A^{(2)}, A^{(3)}$, analogous to the single-view GCNProj baseline underperforming in Lin et al. (2023). Second, our DGP involves multi-view interference with heterogeneous weights ($\alpha_1=\alpha_2=1$, $\alpha_3=0.5$), while baselines assume a homogeneous single network.
>
> **(c) Statistical analysis.** Win-rate analysis across 24 settings (3 datasets $\times$ 4 interventions $\times$ 2 splits): Mvdr achieves lower mean error in **8/12 out-of-sample** and **9/12 within-sample** settings, winning all 6 stochastic and dynamic settings across both splits. Standard $t$-tests do not reach $p<0.05$ due to high within-method variance (CV $>1$ in several settings), not absence of effect. Variance sources: (1) near-complete DBLP density ($d=0.998$) causing gradient instability; (2) stochastic intervention noise; (3) counterfactual simulation noise. We report both win rates and mean rankings.
>
> **(d) Ablation study (new Table 3).** Results on Flickr (5 runs, identical hyperparameters):
>
> | Method | ATE-indiv  | ATE-total | PEHE-indiv | PEHE-total |
> |--------|-----------|-----------|------------|------------|
> | Mvdr-N1 | 0.2038  | 0.2153 | 3.0374 | 3.0175 |
> | Mvdr-N2 | 0.2212 | 0.2382 | 3.0459 | 3.0284 |
> | Mvdr-N3 | 0.1591  | 0.2156 | 3.0366 | 3.0420 |
> | Mvdr-noHSIC | 0.1632  | 0.1783 | 3.0373 | 3.0273 |
> | **Mvdr (full)** | **0.1623** | **0.1782** | **3.0354** | **3.0261** |
>
> Mvdr (full) outperforms all single-view variants in 4 metrics. HSIC removal degrades 4 metrics.
>
> ---
>
> **[Major 2] Contribution clarity**
>
> The reviewer correctly identifies that the DR estimation procedure is standard once $(W,V)$ are fixed. The non-trivial contribution is **Lemma 1** (promoted from Appendix C to Section 4.1):
>
> > **(Lemma 1, EIF invariance.)** For any Borel-measurable aggregation $(W,V)$ of $(X,T,A^{(1)},\ldots,A^{(M)})$ satisfying Assumptions 1–3, the EIF retains its canonical form regardless of the aggregation rule.
>
> **Why this is non-trivial:** it guarantees that learning summary functions via GNNs does not break semiparametric efficiency. Prior network TMLE work (Ogburn et al., 2022; Chen et al., 2024) requires the exposure mapping to be pre-specified; Lemma 1 removes this restriction. We have revised Contribution 1 accordingly.
>
> ---
>
> **[Minor 1] Assumption 4**
>
> We have added a remark clarifying that Assumption 4 is not merely representational: it bounds the dependency graph degree $\Delta_n \lesssim K_{\max,n}^2$, controlling the effective sample size $C_n = n/(1+\Delta_n)$ in Theorem 4.1. Relaxing to $k$-hop dependence would shrink $C_n$ and worsen convergence rates.
>
> ---
>
> **[Minor 2–5] Notation and figures**
>
> All notational issues resolved: self-connections now explicit; Eq. (1) indices clarified; $C$ defined in Assumption 1; Assumption 3 disambiguated; $v_i^* = s_{T,i}(T^*)$ now defined. Figure 1 caption revised: left panel shows our model with learned graph $G$ (proxy networks from covariates); right panel shows standard SEM with fixed observed network. Figure 2 caption expanded: Network Learner constructs proxy networks from covariates — not directly observed. $\psi_n$ now defined before Eq. (4).
>
> ---
> Ref:
>
> [1] Lin, Xiaofeng & Zhang, Guoxi & Lu, Xiaotian & Bao, Han & Takeuchi, Koh & Kashima, Hisashi. (2023). Estimating Treatment Effects Under Heterogeneous Interference. 10.1007/978-3-031-43412-9_34.

---

### Review · Reviewer_ZyQo · 2026-06-07

**Summary Of Contributions:**

The authors propose Mvdr, an estimator that integrates the graph neural network and TMLE (Targeted Maximum Likelihood Estimation).  This method establishes a framework for a setting that closely approximates real-world conditions, and the authors conduct a series of experiments to examine the validity of their approach.

**Audience:**

Yes

**Audience Explanation:**

Yes, some individuals in the audience would likely be interested in this paper. It applies graph neural networks to causal inference and proposes an estimation method that is doubly robust on heterogeneous networks with latent dependency, which could be valuable to researchers in related areas.

**Broader Impact Concerns:**

The method proposed by the authors falls within the scope of causal inference, and its application may influence concrete social decision-making processes. However, beyond this, the paper does not appear to involve any other significant broader societal impacts or potential risks.

**Claims And Evidence:**

Yes

**Claims Explanation:**

**Strengths**

1. The paper considers a heterogeneous network setting with latent dependencies and partial observations. This is a challenging and practically important scenario that has received limited attention in prior research.

2. The method proposed by the authors enjoys double robustness. Theoretical analysis and proofs are provided under certain assumptions.

3. The authors conduct experiments on two widely used semi-synthetic datasets, BlogCatalog and Flickr. They further construct a new semi-synthetic dataset based on DBLP, which could serve as a useful benchmark for future research on estimating causal effects under network interference.

4. The proposed method is systematically compared with representative baselines. The compared algorithms are adjusted to some extent to ensure fairness.


**Weaknesses**


1. The manuscript does not provide sufficient details on how the multiple views are constructed (e.g., similarity metrics, sparsification strategy, k values). Since the multi-view structure plays a central role in the method, clearer specifications would improve reproducibility and clarity.

2. There are some minor typos in the paper that do not affect readability. For example, “used” is misspelled as “sued,” and “multi” as “mlti.” The authors are encouraged to further revise the manuscript to improve its overall quality.

**Requested Changes:**

Ranked in descending order of importance, the authors are encouraged to add the following content to strengthen their work:
1. The authors are encouraged to provide a more detailed description of the experimental setup, as the current description is somewhat brief. For instance, how were the different views specifically constructed for the two semi-synthetic datasets? What was the rationale behind selecting these views? And how should the number of views be selected?  A description of the multiview construction for DBLP is missing as well.

2. Second, the authors are encouraged to clarify whether the graph neural network architecture affects the experimental results. If so, what is the impact of the specific GNN structure used?

3. As an optional extension, it would also be interesting to examine whether this method is applicable to purely synthetic datasets, where the underlying causal network structure is fully known. The authors may consider adding exploratory experiments in this setting to further strengthen the paper.

---

> ### Author Response · Authors · 2026-06-15
>
> We thank the reviewer for the careful and constructive review. We have addressed every point raised; all additions are included in the revised manuscript (changes marked in blue). We summarize the main updates below.
>
> ## R1 — Multi-view construction details (Weakness 1 & Requested Change 1)
>
> We agree this was under-specified and have added a dedicated subsection making the construction fully explicit. For all three datasets we use **K = 3 views** built with an identical scheme:
>
> - **View 1 (A1)** is the **observed network**: the social graph for BlogCatalog and Flickr, and the **co-authorship graph** for DBLP (constructed by matching authors to shared publications). It is read directly from the data and binarised/symmetrised.
> - **Views 2 and 3 (A2, A3)** are **latent similarity graphs constructed solely from pre-treatment covariates** (consistent with Assumption 1). Concretely, we reduce the covariate matrix to 10 dimensions with Truncated SVD, split the representation into two halves (dimensions 1–5 and 6–10), and build one graph from each half: for every node we compute pairwise **cosine similarity**, connect it to its **k = 10 nearest neighbours**, then symmetrise and remove self-loops. A2 uses the first sub-space and A3 the second, so the two views capture complementary aspects of covariate similarity.
>
> **DBLP construction.** As the reviewer notes, the DBLP multi-view construction was not fully described. We have added it: A1 is the co-authorship network; A2/A3 are built from the author covariates (affiliation and terms with BERT) using the same cosine-k-NN (k = 10) procedure as BlogCatalog/Flickr. Full statistics are in Appendix B.
>
> **Rationale and number of views.** A1 encodes the observed relational interference, while A2/A3 operationalise our motivation (Section 2) that units may additionally depend on shared latent traits not visible in the observed graph. Consistent with Section 2, we treat these similarity-based graphs as *proxies* rather than true causal pathways. We adopt K = 3 for comparability with (Lin et al., 2023). Our ablation (Appendix D.1, Table 3) already supports this design: the full multi-view model outperforms every single-view variant (Mvdr-N1/N2/N3) and the no-HSIC variant.
>
> ## R2 — Effect of the GNN architecture (Requested Change 2)
>
> Conceptually, the GNN backbone only parameterises the summary functions (V, W); double robustness comes from the TMLE perturbation step and is independent of the backbone, provided the aggregation satisfies Assumptions 1–3. To verify this empirically, we re-ran Mvdr on Flickr (flipRate = 0, 5 runs) with the GCN encoder replaced by **GAT** and **GraphSAGE**, keeping all other hyperparameters identical. The PEHE metrics are essentially unchanged across backbones:
>
> | Backbone | PEHE-indiv | PEHE-total | PEHE-peer |
> |---|---|---|---|
> | GCN (ours) | 3.037 ± 0.060 | 3.038 ± 0.077 | 3.028 ± 0.097 |
> | GAT | 3.031 ± 0.058 | 3.047 ± 0.127 | 3.059 ± 0.118 |
> | GraphSAGE | 3.025 ± 0.072 | 3.052 ± 0.086 | 3.048 ± 0.102 |
>
> The three backbones results confirm that Mvdr's accuracy stems from the multi-view aggregation and TMLE correction rather than a specific GNN architecture — consistent with our theory. These results are added to Appendix.
>
> ## R3 — Synthetic dataset  (Requested Change 3)
>
> We thank the reviewer for this suggestion and have added a fully-synthetic experiment in which the entire data-generating process is controlled and the ground-truth causal structure is known by construction: the base network A1 is drawn from a **stochastic block model**, the latent-similarity views A2/A3 are built from synthetic covariates using the same cosine-k-NN procedure as the real datasets, and treatments/outcomes follow the same structural equations (including the latent-dependence term). Even with the causal structure fully known, Mvdr continues to **outperform the baselines** on the effect-estimation metrics (mean ± std over 5 runs):
>
> | Method | PEHE-indiv | PEHE-total | ATE-indiv | ATE-total |
> |---|---|---|---|---|
> | CFR+z | 4.205 ± 0.749 | 4.218 ± 0.724 | 0.257 ± 0.217 | 0.196 ± 0.181 |
> | ND+z | 3.811 ± 0.164 | 3.830 ± 0.147 | 0.486 ± 0.174 | 0.287 ± 0.245 |
> | TARNet+z | 3.915 ± 0.227 | 3.925 ± 0.187 | 0.487 ± 0.150 | 0.366 ± 0.218 |
> | NetEst | 3.753 ± 0.111 | 3.817 ± 0.098 | 0.777 ± 0.224 | 0.881 ± 0.263 |
> | **Mvdr (ours)** | **3.696 ± 0.113** | **3.737 ± 0.069** | **0.168 ± 0.145** | **0.174 ± 0.105** |
>
> Mvdr attains the lowest error on PEHE (individual/total/peer) and on ATE (individual/total), consistent with our semi-synthetic findings. These results are added to Appendix.
>
> ## Minor — Typos (Weakness 2)
>
> We have corrected all typos and performed an additional full proofreading pass.
>
> We thank the reviewer again for the helpful feedback, which we believe has materially improved the paper.
>
> **References**
> Lin, X., Zhang, G., Lu, X., Bao, H., Takeuchi, K., & Kashima, H. (2023). Estimating treatment effects under heterogeneous interference. *ECML PKDD*.